# Molecular basis for bacterial *N*-glycosylation by a soluble HMW1C-like *N*-glycosyltransferase

Beatriz Piniello[1], Javier Macías-León[2], Shun Miyazaki[3], Ana García-García [2], Ismael Compañón[4], Mattia Ghirardello [4], Víctor Taleb[2], Billy Veloz[2], Francisco Corzana [4], Atsushi Miyagawa [3], Carme Rovira [1,5] ✉ & Ramon Hurtado-Guerrero [2,6,7] ✉

Soluble HMW1C-like *N*-glycosyltransferases (NGTs) catalyze the glycosylation of Asn residues in proteins, a process fundamental for bacterial auto-aggregation, adhesion and pathogenicity. However, our understanding of their molecular mechanisms is hindered by the lack of structures of enzymatic complexes. Here, we report structures of binary and ternary NGT complexes of *Aggregatibacter aphrophilus* NGT (*Aa*NGT), revealing an essential dyad of basic/acidic residues located in the N-terminal all α-domain (AAD) that intimately recognizes the Thr residue within the conserved motif $Asn^0$-$X^{+1}$-Ser/$Thr^{+2}$. Poor substrates and inhibitors such as UDP-galactose and UDP-glucose mimetics adopt non-productive conformations, decreasing or impeding catalysis. QM/MM simulations rationalize these results, showing that *Aa*NGT follows a $S_N2$ reaction mechanism in which the acceptor asparagine uses its imidic form for catalysis and the UDP-glucose phosphate group acts as a general base. These findings provide key insights into the mechanism of NGTs and will facilitate the design of structure-based inhibitors to treat diseases caused by non-typeable *H. influenzae* or other Gram-negative bacteria.

The attachment of sugar molecules to asparagine residues in proteins, also known as *N*-glycosylation, is a widely occurring post-translational modification found in most eukaryotes[1] and some prokaryotes[2,3]. It has been reported that more than 7000 human proteins are *N*-glycosylated[4]. These *N*-glycans play essential roles in cellular function, mostly in eukaryotes, and are involved in processes such as protein folding and stability, protein trafficking, and signal transduction, playing a major role in health and disease[5,6]. Together with protein *O*-glycosylation, *N*-glycosylation is present in most approved or preclinical protein therapeutics[7–10]. It affects immunogenicity[11] and potency[12], motivating the close study of glycosylation pathways and glycosylation mechanisms[5,13].

*N*-glycosylation is initiated by the membrane-bound oligosacharyltransferase enzyme (OST), which attaches a preassembled oligosaccharide to Asn residues in nascent glycoproteins. OST is a complex oligomeric enzyme in proteins of animals, plants, and fungi[14], whereas it is a monomeric enzyme in bacteria, archaea, and protozoa[15,16]. It was discovered in 2003[17] that bacteria are able to perform a simpler version

¹Departament de Química Inorgànica i Orgànica (Secció de Química Orgànica) and Institut de Química Teòrica i Computacional (IQTCUB), Universitat de Barcelona, Martí i Franquès 1, 08028 Barcelona, Spain. ²Institute of Biocomputation and Physics of Complex Systems, University of Zaragoza, Mariano Esquillor s/n, Campus Rio Ebro, Edificio I+D, Zaragoza, Spain. ³Department of Life Science and Applied Chemistry, Nagoya Institute of Technology, Gokiso-cho, Showa-ku, Nagoya 466-8555, Japan. ⁴Departamento de Química, Universidad de La Rioja, Centro de Investigación en Síntesis Química, E–26006 Logroño, Spain. ⁵Institució Catalana de Recerca i Estudis Avançats (ICREA), Passeig Lluís Companys 23, 08010 Barcelona, Spain. ⁶Copenhagen Center for Glycomics, Department of Cellular and Molecular Medicine, University of Copenhagen, Copenhagen, Denmark. ⁷Fundación ARAID, 50018 Zaragoza, Spain. ✉ e-mail: c.rovira@ub.edu; rhurtado@bifi.es

of N-glycosylation in which a single monosaccharide or a disaccharide, rather than a complex oligosaccharide, is attached to specific asparagine residues. This process is catalyzed by soluble N-glycosyltransferase enzymes (NGTs). Bacterial N-glycosylation, which is either accomplished by OSTs or NGTs, is important for bacterial survival, adhesion, autoaggregation, and pathogenicity[6,16–18].

The first known example of an NGT was discovered in non-typeable *Haemophilus influenzae*[17]. This enzyme, termed HMW1C, was demonstrated to be capable of adding mono- and di-hexose units onto Asn residues of proteins, such as HMW1 adhesin[18]. Similar enzymes were later discovered in many other Gram-negative bacteria, including *Actinobacillus pleuropneumoniae*[19], *Haemophilus ducreyi* and *Mannheimia haemolytica*[20], *Yersinia pestis* and *Escherichia coli*[10], *Aggregatibacter aphrophilus*[11], *Kingella kingae*, and *Bibersteinia trehalosi*[21].

Although OSTs and NGTs differ in the type of donor substrate (lipid-bound sugars for OSTs versus nucleotide-bound sugars for NGTs), they are both inverting GTs, i.e. they catalyze the formation of a glycosidic bond with inversion of configuration of the donor anomeric carbon[22,23]. They also share the $Asn^0\text{-}X^{+1}(X \neq P)\text{-}Ser/Thr^{+2}$ conserved sequence motif in their protein acceptor substrates[24] (Fig. 1a). The numbering for the sequon follows a specific convention. The acceptor Asn residue is assigned the number 0, and the subsequent amino acids are then numbered in a positive manner. However, the architecture of OSTs and NGTs differ significantly. Structural studies have shown that bacterial and eukaryotic OST catalytic domains are multi-spanning membrane glycosyltransferases (GTs) that adopt a GT-C fold[15,25,26]. In contrast, NGTs are two-domain enzymes that consists of an N-terminal all-α domain (AAD) and a C-terminal catalytic domain that adopts a GT-B fold, as shown for *Actinobacillus pleuropneumoniae* NGT (*Ap*NGT)[23]. In addition, while the OSTs require a metal, preferably $Mn^{+2}$, to catalyze the reaction, no metal is necessary for the NGTs[13,15,16,18].

Despite previous structural studies on the free form of *Ap*NGT and in complex with UDP, the peptide substrate recognition and the catalytic mechanism of either OSTs or NGTs enzymes remain unknown[27]. Most inverting GTs follow a mechanism in which a catalytic base deprotonates the incoming nucleophile of the acceptor (here Asn; Fig. 1a). However, previous attempts to gain a better comprehension of NGT catalytic mechanism via site-directed mutagenesis of active site residues have been unsuccessful, raising questions regarding the identity of the catalytic base that deprotonates the Asn acceptor, which remains unidentified[28].

Here, we have employed a multidisciplinary approach involving structural biology, synthetic chemical biology, kinetic experiments, and computational techniques to uncover the recognition of substrates and the catalytic mechanism of *Aggregatibacter aphrophilus* NGT (*Aa*NGT), an inverting GT that is an orthologue of *Ap*NGT. Our results indicate that Thr residues at positions +2 and +3 are the main determinants of peptide-NGT interactions, while poor donor substrates such as UDP-galactose (UDP-Gal), and inhibitors such as UDP-Glc mimetics, adopt unproductive conformations that compete with the peptide substrate, resulting in reduced or impeded catalysis, respectively. QM/MM metadynamics simulations show that *Aa*NGT follows a concerted single-displacement $S_N2$ mechanism in which the acceptor Asn attacks the sugar donor anomeric carbon via the imidic form and the α-phosphate of UDP-Glc acts as the catalytic base.

## Results

### Kinetic and binding of *Aa*NGT against a model peptide and sugar nucleotides

To perform biophysical experiments using *Aa*NGT, we designed a full-length construct that was expressed in *E. coli* (residues M1-I621; Supplementary Fig. 1 and "Methods" section). To evaluate its activity, we synthesized a peptide containing the GNWT motif that was found to be robustly glucosylated by other NGTs in the context of a longer peptide[20]. Additionally, our peptide contained a Thr residue at +3,

which was demonstrated to improve glucosylation, and a Phe residue at −2, which was found to be tolerated in terms of glycosylation[20]. *Aa*NGT showed a hyperbolic profile in the presence of variable concentrations of UDP-Glc or FGNWTT (Fig. 1b). The $K_m$s for UDP-Glc and FGNWTT were determined to be $90 \pm 30$ and $79 \pm 11$ μM, respectively, and the $k_{cat}$ values were $20 \pm 2$ and $16 \pm 1$ min$^{-1}$, respectively (Fig. 1b and Supplementary Table 1). These values are in accordance with previously reported $k_{cat}$ values for other NGTs, which range from 18 to 300 min$^{-1}$ depending on the peptide sequences[20,28]. To investigate other sugar nucleotides, we compared the initial velocities of *Aa*NGT using UDP-Glc, UDP-Gal, and UDP-Glc mimetics (UDP-2F-Glc and UDP-5S-Glc), being the latest prepared according to our previous methodologies[29] (see Supplementary Figs. 3 and 4 and "Methods" section). We used 500 μM concentrations of the sugar nucleotides and the peptide (saturated conditions found for UDP-Glc and the peptide), and found that the initial velocity with UDP-Gal was ~40-fold lower than that of UDP-Glc and completely inactive with either UDP-2F-Glc or UDP-5S-Glc (Fig. 1c), implying that we could potentially trap co-crystals of *Aa*NGT complexed to UDP-Glc mimetics and the peptide. Note that similar results with the sugar nucleotides were previously found for other NGTs and *Aa*NGT[20,28,30]. We then performed isothermal titration calorimetry (ITC) experiments with the sugar nucleotides, UDP and the peptide. While reasonable titration curves were obtained with the sugar nucleotides and UDP (Fig. 1d and Supplementary Fig. 2), the peptide unfortunately precipitated during the ITC experiment, which impeded obtaining useful data. The results of our ITC experiments demonstrated that the product of the reaction, UDP, binds to *Aa*NGT ~ 3–5.8-fold and 18-fold better than UDP-Glc mimetics and UDP-Gal, respectively (Fig. 1e and Supplementary Table 2). Analysis of the thermodynamic parameters indicated that the interaction of the enzyme with UDP and UDP-Gal is enthalpically favored, while that with UDP-2F-Glc and UDP-5S-Glc is entropically favored. These findings suggest that the binding modes and/or interactions of the ligand-protein complexes vary depending on the nucleotide (Supplementary Table 2).

Overall, our results of kinetic and binding experiments indicate that UDP-Glc is a preferred substrate of *Aa*NGT compared to UDP-Gal, whereas UDP-Glc mimetics act as inhibitors of the enzyme. In addition, the data suggest that the enzyme binds worse to the sugar nucleotides than UDP.

### Architecture of binary and ternary complexes of *Aa*NGT

Having characterized *Aa*NGT with different sugar nucleotides, UDP, and the peptide FGNWTT, we obtained $P2_12_12_1$ crystals of *Aa*NGT in complex with UDP-Gal, UDP-2F-Glc, and UDP plus the peptide (Table 1). Diffraction was poor for crystals obtained in the presence of UDP-5S-Glc, thus precluding obtaining useful data with this sugar nucleotide. Attempts to obtain a ternary complex that resembles a Michaelis complex by combining the UDP-2F-Glc or UDP-Gal with the peptide yielded a complex without peptide (see below for an explanation). These crystal structures have resolutions between 1.76 and 2.80 Å, and contain two molecules per asymmetric unit (Table 1). Root mean square deviation (RMSD) values of ~0.32 Å and 0.54 Å were computed between chains A and B of the asymmetric unit for the enzyme complex with UDP-sugar (UDP-Gal/UDP-2F-Glc) and UDP-peptide, respectively. This reveals that the two *Aa*NGT molecules are slightly more different in the complex with UDP and the peptide compared to the complexes with UDP-sugar. Analysis of the *Aa*NGT structure with the DALI server[31] revealed structural homology to three top hits: *Ap*NGT (PDB entries 3Q3E, 3Q3I, and 3Q3H), the protein O-fucosyltransferase SPINDLY from *Arabidopsis thaliana* (PDB entry 7Y4I) and the *Xanthomonas campestris* putative O-GlcNAc transferase, OGT (PDB entries 2VSY, 2JLB and 2VSN). As expected, *Ap*NGT has the lowest RMSD of 1.4 Å on 618 aligned residues, due to the high structural homology between both NGTs (Fig. 2a and Supplementary Fig. 1).

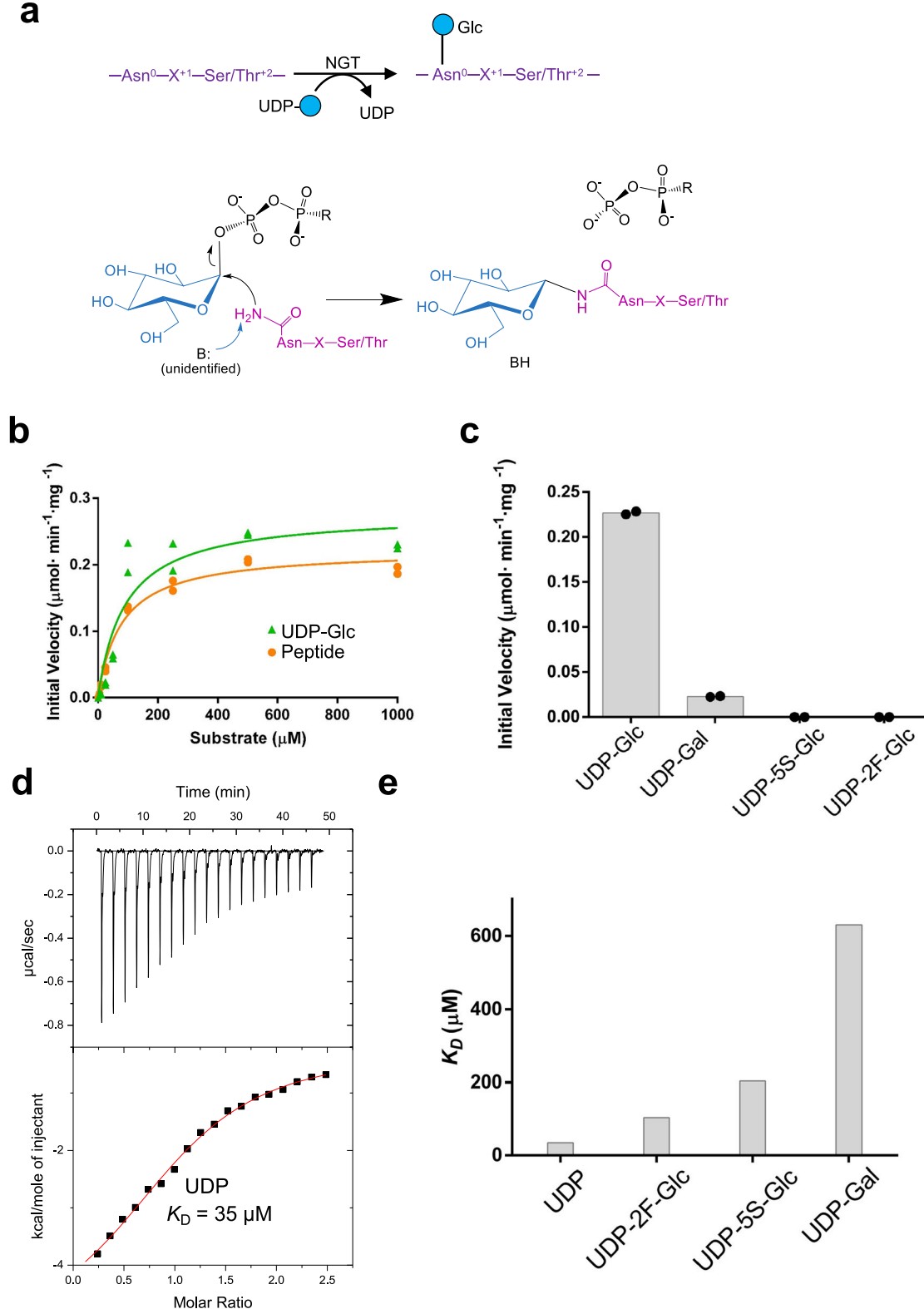

**Fig. 1 | Enzyme kinetics and ITC experiments of *Aa*NGT. a** Scheme illustrating the general reaction of NGTs. **b** Glycosylation kinetics of *Aa*NGT using UDP-Glc and the peptide FGNWTT. **c** Plots comparing the initial velocities of *Aa*NGT in the presence of different sugar nucleotides. Additional kinetic data are given in Supplementary Table 1. All experiments were obtained in duplicate (*n* = 2 independent experiments). **d** ITC data for the binding of UDP to *Aa*NGT. Top: raw thermogram (thermal power versus time). Bottom: binding isotherm (normalized heats versus molar ratio). The experiment was repeated at least 2 times independently with similar results, and one representative plot for each experiment is shown. **e** Graph depicting the $K_d$s for the nucleotides (see Supplementary Table 2 for all ITC data). Source data are provided as a Source Data file.

**Table 1 | Data collection and refinement statistics**

| | AaNGT-UDP-Gal | AaNGT-UDP-2F-Glc | AaNGT-UDP-peptide |
|---|---|---|---|
| Data collection | | | |
| Space group | P2₁2₁2₁ | P2₁2₁2₁ | P2₁2₁2₁ |
| Wavelength (Å) | 0.9792 | 0.9792 | 0.9792 |
| Cell dimensions | | | |
| a, b, c (Å) | 47.34, 113.33, 260.11 | 47.07, 114.00, 258.01 | 47.03, 111.48, 256.37 |
| α, β, γ (°) | 90, 90, 90 | 90, 90, 90 | 90, 90, 90 |
| Number of protein molecules per asymmetric unit | 2 | 2 | 2 |
| Resolution (Å) | 20–1.76 (1.86–1.76)ᵃ | 258.01–2.73 (2.88–2.73)ᵃ | 20–2.80 (2.95–2.80)ᵃ |
| $R_{merge}$ | 0.065 (1.796) | 0.104 (2.173) | 0.177 (2.049) |
| $R_{pim}$ | 0.025 (0.691) | 0.044 (0.914) | 0.063 (0.740) |
| Mn(I) half-set correlation CC(1/2) | 0.999 (0.430) | 0.998 (0.458) | 0.996 (0.352) |
| $I / \sigma I$ | 14.2 (1.2) | 8.7 (0.9) | 8.2 (2.1) |
| Completeness (%) | 99.9 (100) | 100 (100) | 99.7 (100) |
| Redundancy | 7.7 (7.7) | 6.6 (6.6) | 8.4 (8.1) |
| Total number of reflections | 1074677 | 250180 | 286978 |
| Total number unique reflections | 139852 | 38131 | 34243 |
| Refinement | | | |
| Resolution (Å) | 1.76 | 2.73 | 2.80 |
| $R_{work}/R_{free}$ | 0.173 (0.2096) | 0.197/0.245 | 0.178/0.237 |
| No. atoms | | | |
| Protein | 9958 | 9966 | 9919 |
| Peptide | – | – | 79 |
| UDP | – | 25 | 50 |
| Ethylenglycol | 102 | – | – |
| UDP-Gal | 36 | – | – |
| UDP-2F-Glc | – | 36 | – |
| Waters | 787 | 8 | 5 |
| B-factors (Å²) | | | |
| Protein | 40.66 | 111.37 | 88.56 |
| Peptide | – | – | 109.35 |
| UDP | – | 144.216 | 86.30 |
| Ethylenglycol | 57.15 | – | – |
| UDP-Gal | 47.30 | – | – |
| UDP-2F-Glc | – | 124.72 | – |
| Waters | 45.87 | 74.16 | 60.51 |
| R.m.s. deviations | | | |
| Bond lengths (Å) | 0.0138 | 0.0061 | 0.0095 |
| Bond angles (°) | 1.8771 | 1.396 | 1.6565 |

One crystal was used to determine the crystal structure.
ᵃValues in parentheses are for highest-resolution shell.

On the contrary, the second and third top hits superimposed less well with AaNGT (RMSDs of ~3.4 and ~4.8 Å between SPINDLY and AaNGT, and XcOGT and AaNGT crystal structures, respectively; the superimposed residues ranged from 453 to 467 residues). This is likely due to their different GT-B fold catalytic domains, as these GTs bind to different sugar nucleotides and glycosylate different protein substrates[32–34]. Furthermore, NGTs possess a distinct N-terminal AAD domain, while XcOGT and SPINDLY contain N-terminal TPRs, which differ in fold from the AAD domain.

The heart-shaped structure of AaNGT (Fig. 2a), either complexed to UDP-Gal/UDP-2F-Glc or UDP-peptide, shows the N-terminal AAD domain and C-terminal GT-B fold catalytic domain, which is composed of an N- and C-Rossman fold subdomains. All ligands were well resolved, except for the Phe1⁻² side chain of the peptide, which did not display any density (Fig. 2b), likely explaining why Phe at site -2 is only tolerated by NGTs without any preference[20]. A surface representation of the ternary complex reveals that UDP is partly buried and mainly recognized by the C-Rossmann fold subdomain while the peptide is more solvent exposed and tethered mostly by the N-Rossmann fold subdomain and the AAD domain (Fig. 2c).

**The sugar nucleotide binding site of AaNGT**
The AaNGT substrate binding site is formed by the sugar nucleotide and the peptide binding sites (Fig. 3a). The uracil moiety of UDP/UDP-Gal/UDP-2F-Glc is stabilized by CH–π interactions with His494 and Tyr500, alongside hydrogen bonds to the side chain and backbone of Ser495. The ribose moiety is recognized in all three complexes by hydrogen bonds to Asp524. Additionally, interactions with Arg280 are observed in the ternary complex. The UDP pyrophosphate of the ternary complex (AaNGT + UDP + peptide) exhibits more interactions than in the binary complexes, with Thr519, Asn520, and Gly521 backbones, and Ser277, Thr437, Lys440, and Asn520 side chains contributing to this. In contrast, the UDP-Gal and UDP-2F-Glc pyrophosphates are only recognized by the side chains of Lys440 and Asn520. The sugar molecules interact with the side chains of His276 and Ser277. Furthermore, the Gal moiety specifically interacts with the Gly369 backbone, while the 2F-Glc moiety interacts with the side chain of His370 (Fig. 3a). Therefore, these more extensive interactions of UDP with the enzyme could explain the higher affinity of UDP compared to the sugar nucleotides (Figs. 1e and 3a). Furthermore, superposition of the nucleotide structures reveals significant differences in the uracil moiety for UDP versus UDP-Gal/UDP-2F-Glc (specifically, the β-phosphate orientation differs significantly), as well as in the ribose and pyrophosphate between all structures and the sugar moieties (Fig. 3b). This suggests that the binding of the nucleotides to the enzyme is flexible and dynamic, which could contribute to the catalytic pathway of NGTs.

**The peptide binding site of AaNGT**
Multiple hydrogen bond interactions are observed at the peptide binding site in the ternary complex (AaNGT-peptide-UDP). These interactions include the backbone atoms of Phe1⁻² and Ser277, the backbone of Asn3⁰ and the UDP α-phosphate, the side chain of Asn3⁰ and Gly369, the backbone of Thr5⁺² and the Arg177 side chain, the Thr5⁺² and Asp215 side chains, the Thr6⁺³ backbone and the Arg177 side chain, and the Thr6⁺³ and His214 side chains (Fig. 3a). The structural data demonstrate that the key residues involved in recognition of the peptide are Arg177, His214, and Asp215 from the AAD domain. Notably, Thr5⁺², which is part of the Asn⁰-X⁺¹-Ser/Thr⁺² sequence motif, is differently recognized by OSTs, which instead use the WWD sequence to recognize it. This indicates that the mechanisms employed to recognize peptide substrates by NGTs and OSTs differ. Furthermore, the interactions between Thr6⁺³ and Arg177/His214 provide an explanation for the previously reported kinetic data on other NGTs, which showed that Thr at position +3 improves the kinetics against peptides with this particular residue[20].

CH–π interactions between the peptide and the protein were also observed, in particular those involving Trp4⁺¹ and the methylene group of Gln468 (AaNGT). The latter residue is located at the peptide binding site and its importance for enzyme activity was previously demonstrated. In fact, the equivalent residue in ApNGT (Gln469) was found to be deleterious for NGT activity, whereas its mutation to Ala significantly improved NGT activity[35]. The superposition of all ligands for all three structures (AaNGT/UDP, AaNGT/UDP-Gal, and AaNGT/UDP-

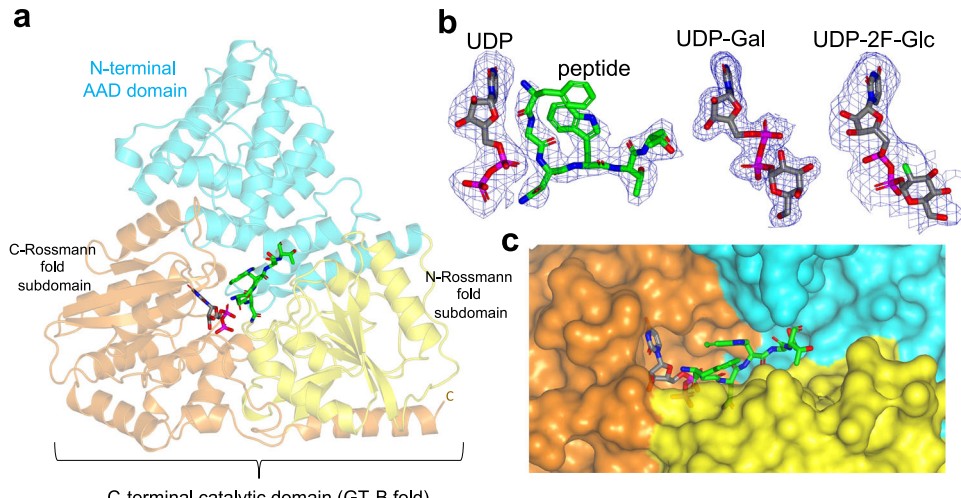

**Fig. 2 | Crystal structures of *Aa*NGT complexed to UDP-Gal, UDP-2F-Glc, and UDP-FGNWTT. a** Ribbon structure of the *Aa*NGT complexed to UDP and FG<u>N</u>WTT. The N-terminal AAD, the N-terminal Rossmann and C-terminal Rossmann fold subdomains are colored in cyan, yellow and orange, respectively. The UDP nucleotide is depicted with gray carbon atoms whereas the peptide is shown as green carbon atoms. In **b** Close-up view of the active site showing the bound UDP and FG<u>N</u>WTT, UDP-Gal, and UDP-2F-Glc in the different complexes. Electron density maps are Fo−Fc (blue) contoured at 2.2 σ for all ligands. Except for the first N-terminal residue (Phe1$^{-2}$) of FG<u>N</u>WTT, the density for the peptide and nucleotides were well defined. **c** A close-up view of the surface representation of the *Aa*NGT active site is displayed, with the same colors as in panel **a**.

2F-Glc) clearly shows that the sugar moieties of UDP-Gal and UDP-2F-Glc collide with Gly2$^{-1}$ and Asn3$^{0}$ (Fig. 3b). This suggests that the sugar nucleotides adopt unproductive conformations in their respective enzyme complexes, which could explain why we were unable to get a ternary complex resembling a Michaelis complex at a structural level. Furthermore, this could also explain why UDP-Gal is a poor substrate, and UDP-Glc mimetics are inhibitors of *Aa*NGT.

To gain insight into the role of the AAD residues of *Aa*NGT involved in peptide recognition, we tested Ala mutations of Arg177, His214, and Asp215 to Ala residues. The resulting mutants were characterized at the in vitro level under the same conditions used for the wild-type enzyme. The results showed that R177A and D215A are inactive while H214A exhibit an 11-fold and 27-fold decrease in activity and catalytic efficiency compared to those of the WT. Additionally, slight variations were observed regarding the $K_{m}$s of the peptide (Fig. 3c and Supplementary Table 1). A triple mutant to Ala residues, as expected, was also inactive (Fig. 3c).

In summary, the structural analysis shows that NGT shares two critical and conserved residues among NGTs (Supplementary Fig. 1), Arg177 and Asp215, in the AAD domain that recognize Thr and likely Ser at the +$2$ position of the Asn$^{0}$-X$^{+1}$-Ser/Thr$^{+2}$ sequence motif. The Asn at the acceptor position $O$ in the sequence motif (Asn3$^{0}$) is located in front of the UDP pyrophosphate, although its side chain does not participate in interactions with any residue of *Aa*NGT that could potentially serve as a catalytic base. Consequently, our ternary complex does not supply any evidence on how the Asn is glycosylated and thus no insights into the NGT catalytic mechanism could be inferred.

## Modeling the Michaelis complex

To get insight into the catalytic mechanism of *Aa*NGT, we turned to computer simulation using the structures determined in the present work. As pointed out above, the Michaelis complex, *i.e.* the complex of *Aa*NGT with UDP-Glc and the peptide acceptor, cannot be reconstructed by structural superposition of the ternary and binary complexes (*Aa*NGT-UDP-Gal and *Aa*NGT-peptide, respectively) due to strong steric clash between the sugar and the peptide (Fig. 3b), which precludes using structural superposition to start Molecular dynamics (MD) simulations. Therefore, we used molecular docking to insert UDP-Glc in the binding site of the *Aa*NGT-peptide binary complex. This

resulted in a structure with no steric clash that provided a very good starting point for MD simulation.

Interestingly, the binding pose of UDP-Glc in the modeled Michaelis complex was found to be remarkably similar to that observed for the UDP-GlcNAc donor in *O*-GlcNAc transferase (OGT). O-GlcNAc transferase is an inverting glycosyltransferase with a GT-B fold highly similar to that of *Aa*NGT. It is noteworthy that despite OGT catalyzing a distinct reaction (*O*-glycosylation), both enzymes are classified in the same CAZy family, GT41, owing to their similarities at the catalytic domain level[36]. MD simulations for up to 400 ns (Supplementary Fig. 5) were performed to further accommodate the UDP-Glc and the acceptor peptide in the active site. As shown in Fig. 4a, the hydroxyl substituent groups of the donor glucose are engaged in hydrogen bond interactions with the side chain of Asn520 and the backbone of Leu368. The interaction of the 2-OH and the β-phosphate group could explain why the enzyme cannot recognize UDP−2F-Glc in a productive manner. Likewise, the interactions between OH3 and OH4 of Glc with Leu368 could explain the poor interaction with UDP-Gal.

The simulations showed that the peptide Asn residue (Asn3$^{0}$) is located on the β face of the donor sugar, opposite to the sugar-phosphate bond, with the amino group close to the Glc anomeric carbon (N-C1 ≈ 3.7 Å, see Fig. 4b). It was previously suggested that the Asn uses a twisted form of the amide group as a prerequisite to enhance the nucleophilicity of the amine group in order to facilitate its attack on the sugar anomeric carbon in OST[15,25,26]. Such twist of the amide was not observed in *Aa*NGT, which shows the amide of the Asn in its most common planar conformation (Fig. 4). The different architectures of the two active sites, in particular the lack of metal-coordinating residues in NGTs, is likely the cause of this difference.

As inferred from the crystal structure complexes, no residue serving as a general catalytic base in the anticipated S$_{N}$2 reaction could be identified. This was attributed to the lack of hydrogen bonding between the amino group of Asn3$^{0}$ and a nearby amino acid that could potentially act as a catalytic base. However, the amino group forms a persistent hydrogen bond interaction with one of the negatively charged oxygen atoms of the α-phosphate (H··O$_{\alpha}$ ≈ 1.8 Å, Fig. 4c). This suggests that the α-phosphate could deprotonate Asn3$^{0}$ during the S$_{N}$2 reaction. In fact, the α-phosphate has been proposed to be the general base in other glycosylation mechanisms, such as the *O*-glycosylation

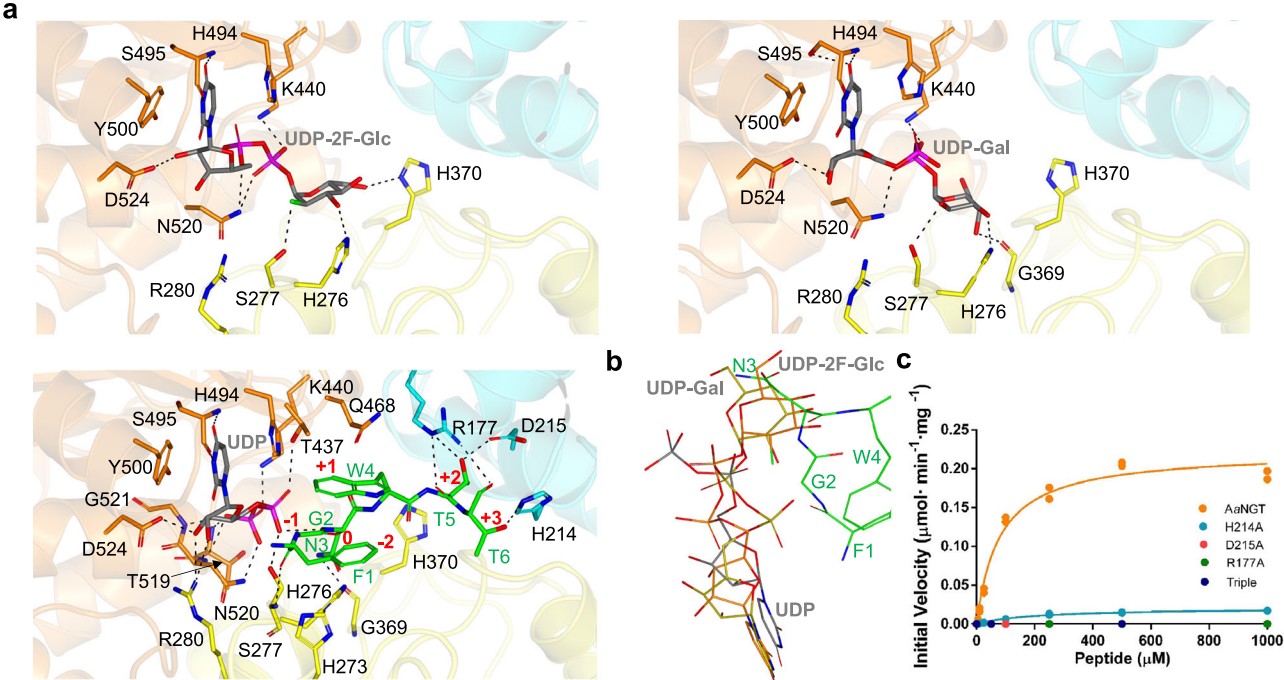

**Fig. 3 | Structural features of the peptide and sugar nucleotide binding sites of _Aa_NGT. a** View of the active sites of _Aa_NGT-UDP-2F-Glc (upper-left panel), _Aa_NGT-UDP-Gal (upper-right panel), and _Aa_NGT-UDP-FGNWTT (lower-left panel) complexes. Residues are colored according to their location in the different domains and subdomains of _Aa_NGT, with the same color scheme as used in Fig. 2a. The nucleotides UDP and the peptide are shown as gray and green carbon atoms, respectively. Hydrogen bond interactions are displayed as dotted black lines.

**b** Superposition of the different ligands with UDP-Gal as yellow carbon/phosphate atoms, UDP-2F-Glc as orange carbon/phosphate atoms, UDP as gray carbon/phosphate atoms, and FGNWTT as green carbon atoms. **c** Glycosylation kinetics of _Aa_NGT and mutants, measured against variable concentrations of the peptide FGNWTT and using a saturated concentration of UDP-Glc. Additional kinetic data are provided in Supplementary Table 1. All experiments were obtained in duplicate ($n = 2$ independent experiments). Source data are provided as a Source Data file.

mechanism of the closely related _O_-GlcNAc transferase (OGT) and the recently investigated plant protein _O_-fucosyltransferase SPINDLY (in both cases, the α-phosphate is presumed to deprotonate the hydroxyl group of Ser/Thr)[34,37]. The possible catalytic base character of the α-phosphate is further supported by NMR and computational experiments that demonstrate that the p$K_a$ of the α-phosphate in UDP, UDP-GlcNAc, and UDP-S-GlcNAc is ~6.5[38,39].

### Modeling the _N_-glycosylation reaction

To model the reaction mechanism of _Aa_NGT, we selected one representative snap-shot of the classical MD simulation and performed quantum mechanics/molecular mechanics (QM/MM) MD simulations (for 5 ps), using a QM region that included most of the donor molecule (Glc and the two phosphate groups), and the side chain of Asn3[0] (44 QM atoms, 106371 MM atoms). The active site remained in a similar configuration as in the previous MD simulations, in which the Asn3[0] amino group is engaged in a hydrogen bond with the α-phosphate. Afterwards, we tried to model the _N_-glycosylation reaction using QM/MM metadynamics, as well as other enhanced-sampling methods (see Methods) previously used to study glycosylation reactions[40–43]. Unfortunately, all our attempts to obtain a feasible reaction pathway failed. Even though the Asn3[0] residue became glycosylated during the simulation (Supplementary Fig. 6), the free energy barrier of the reaction was found to be huge (>50 kcal/mol), indicating that _N_-glycosylation via the amide side chain of Asn3[0] is not feasible. We reasoned that this is due to the particular orientation of Asn3[p] with respect to the sugar donor, which results in an unfavorable stereochemistry for the approach of the nucleophile (<N-C1-O$_P$ ≈ 115°, very far from the optimum value of 180°).

Surprisingly, one of the attempts to model the _N_-glycosylation reaction resulted in tautomerization of the amide group of Asn3[0] via one oxygen atom of the α-phosphate. In other words, one of the amide

protons transferred to O$_\alpha$ and, subsequently, the amide carbonyl abstracted the proton from O$_\alpha$, resulting in the imidic form of Asn3[0]. It was interesting to observe that the N atom of the imidic Asn is better poised for nucleophilic attack than the Michaelis complex with Asn3[0] in the amide form. In particular, the N$_{Asn}$-C1-O1 angle in the imidic Asn increases by ≈ 15° (from 115° to 130°) with respect to the angle in the amide complex (Supplementary Fig. 8). At the same time, the hydrogen atom of the Asn3[0] hydroxyl group forms a hydrogen bond with the α-phosphate, favoring proton transfer. These results made us think that the imidic form of Asn3[0], rather than the most common amidic form, could be operative in the _N_-glycosylation reaction.

To evaluate the possibility of _N_-glycosylation via the imidic form of Asn3[0], we performed QM/MM metadynamics[44] simulations of the chemical reaction, starting from the imide form Asn3[0], which turned out to be stable in the active site (both in MD and QM/MM MD simulations). We used two collective variables corresponding to the main covalent bonds that need to be formed or broken during the reaction: the nucleophilic attack distance (N-C1) and the leaving group distance (C1-O$_P$) (Fig. 5a). During the simulation, the system successfully evolved from the Michaelis complex (_Aa_NGT + UDP-Glc + peptide) to the reaction products (_Aa_NGT + UDP + Glc-peptide), in which Asn3[0] is glycosylated (Fig. 5b, c). The reaction free energy landscape (Fig. 5b) shows a unique transition state (TS), thus it is consistent with a concerted one-step S$_N$2 reaction. The computed free energy barrier (24.9 kcal/mol) is still somewhat higher than the one estimated from the experimental rate constant (18.4 kcal/mol, assuming Transition State Theory)[45], probably due to an imperfect position of the phosphate groups in the initial structures. However, it is similar to the one previously computed for OGT (23.5 kcal/mol)[46]. Most importantly, the free energy barrier is much reduced compared to the one obtained for the reaction via the amide form of Asn3[0] (> 50 kcal/mol), indicating that the reaction occurs preferably via the imidic form of Asn3[0].

**a**

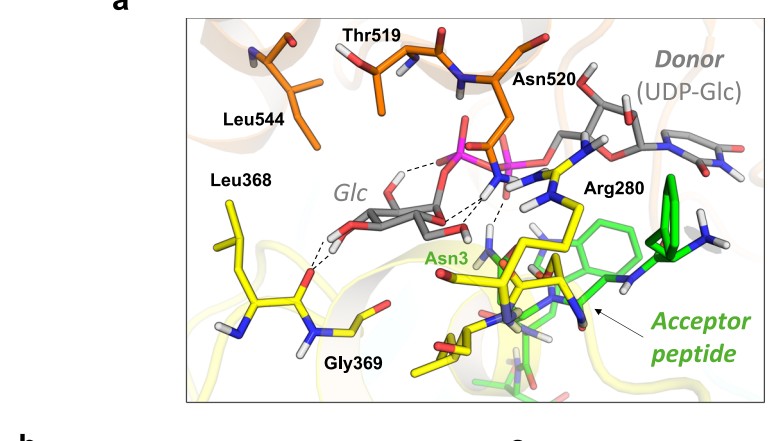

**b**

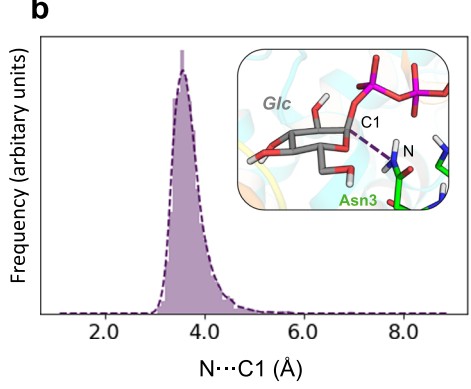

**c**

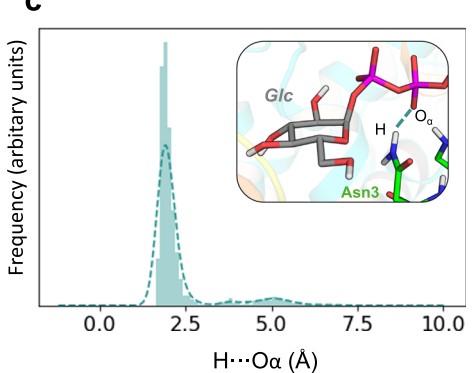

**Fig. 4 | The Michaelis complex of *Aa*NGT. a** Representative snap-shot obtained from classical MD simulations, indicating the hydrogen bond interactions involving the sugar donor and the hydrogen bond between Asn3⁰ and the donor α-phosphate. **b** Distribution of values of the nucleophilic attack distance (N···C1) during the classical MD simulation. **c** Distribution of values of the distance between one H atom of the Asn amide group and its closest phosphate O atom (H···$O_\alpha$) during the classical MD simulation. Plot data in Source Data file.

Representative structures of the MC, transition state (TS), and reaction products (P) are shown in in Fig. 5d. Interestingly, a hydrogen bond between the α-phosphate and Trp4⁺¹ is present at the MC, which probably contributes to position the peptide in the active site. This might explain recent experimental results that show that particular peptide sequences with a Trp at that position improves the glucosylation efficiency[20]. The computed reaction pathway also shows a TS in which the sugar-phosphate bond is being broken (C1-$O_P$ = 2.2 Å) and the sugar-Asn3⁰ bond is being formed (C1-N = 2.1 Å) (Fig. 5c and Supplementary Table 3). Simultaneously, the proton of the imide hydroxyl group is being transferred, leading to the products of the reaction, which is lower in energy with respect to the reactants. Note that the structure of the reaction products does not depend on the tautomerization state of Asn3⁰ at the initial state (MC).

Finally, we sought to elucidate the most likely mechanism of Asn tautomerization in the *Aa*NGT active site. To this end, we performed QM/MM metadynamics simulations of the tautomerization process considering two possible scenarios: tautomerization mediated by the α-phosphate or tautomerization via active site water molecules (Supplementary Fig. 7). In both cases, two collective variables were used to drive the system from the amidic to the imidic form of the Asn3⁰ side chain. Whereas tautomerization via the α-phosphate involves an energy barrier of 29.5 kcal/mol, the energy barrier reduces to 17.5 kcal/mol when Asn3⁰ undergoes tautomerization via water molecules. This indicates that asparagine tautomerization in the active site is feasible and it is mediated by active site water molecules that are properly positioned for proton shuttle.

In summary, our simulations suggest that the *N*-glycosylation reaction in *Aa*NGT, and probably other NGTs, takes place via a S$_N$2 reaction in which the Asn3⁰ in its imidic form attacks the sugar donor anomeric carbon. The nucleophilic attack is assisted by proton transfer to the α-phosphate, which can be considered as the general base of the *N*-glycosylation reaction in *Aa*NGT (Fig. 5e).

## Discussion

*N*-glycosylation is the most common post-translational modification (PTM) of proteins[47]. However, it is intriguing to understand why Nature has chosen the amide group of Asn residues, which is one of the least reactive nucleophiles in proteins, for glycosylation. Although the exact reason for selecting Asn residues for glycosylation remains unknown, we can investigate the mechanistic strategies utilized by various types of GTs to accomplish this PTM, which have arisen throughout evolution and are widespread in eukaryotes and certain bacteria. These GTs include single or complex membrane-bound oligosaccharyl-transferases (OSTs) and soluble *N*-glycosyltransferases (NGTs).

In this study, we adopted a multidisciplinary approach to shed light on the mechanism of *N*-glycosylation catalyzed by NGTs. We focused on *Aa*NGT, an inverting GT that catalyzes the transfer of Glc from UDP-Glc to Asn residues in the consensus sequence Asn-X-Ser/Thr. We provide high-resolution crystal structures for the enzyme in complex with a poor donor substrate (UDP-Gal) and a donor mimic (UDP-2F-Glc), as well as the enzyme in complex with a peptide acceptor (FGNWTT) and UDP, together with kinetic/ITC experiments and QM/MM metadynamics simulations of the reaction mechanism. Our results indicate that Thr residues at positions *+2* and *+3* are the main determinants of peptide-NGT interactions and the AAD domain recognizes these residues. Donor substrates less favorable than UDP-Glc, like UDP-Gal and inhibitors such as UDP-Glc mimetics, adopt unproductive

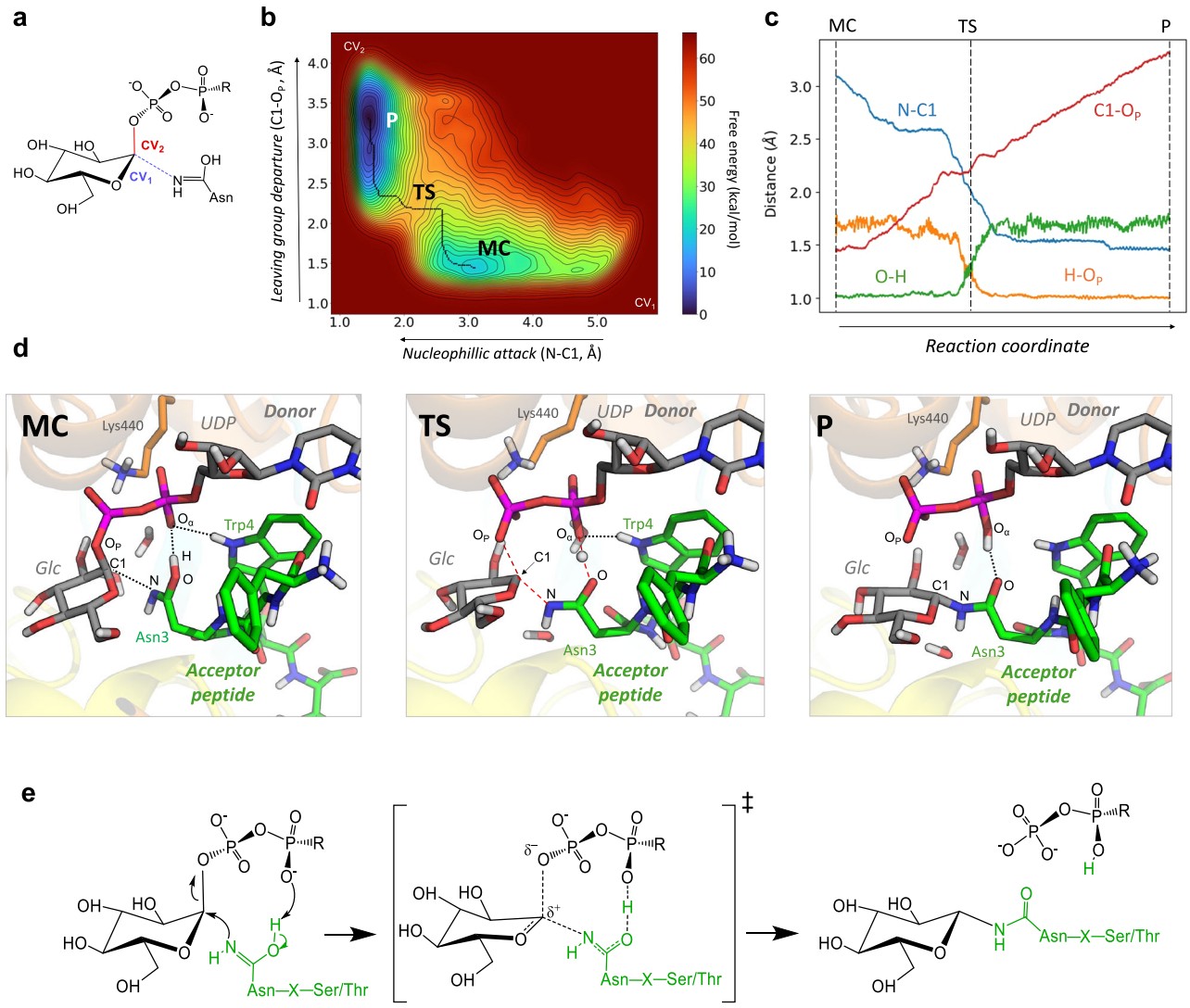

**Fig. 5 | Reaction mechanism of the *N*-glycosylation reaction catalyzed by *Aa*NGT. a** Collective variables used in the QM/MM metadynamics simulation. **b** Computed reaction free energy landscape. MEP computed with MEPSAnd software[76]. FES data provided in the Zenodo repository (https://doi.org/10.5281/zenodo.8081487). **c** Evolution of relevant distances along the reaction coordinate (minimum free energy pathway connecting MC and P). A running average was applied (*n* = 10). Plot data including all values is provided in the Source Data file. **d** Representative structures of the main states along the reaction coordinate. Relevant hydrogen bonds are depicted as dotted lines, whereas covalent bonds being formed or broken are indicated by red dashed lines. **e** Scheme of the reaction mechanism proposed in this work.

conformations that compete with the peptide substrate, ultimately leading to decreased or hindered catalysis. These results are confirmed by MD simulations of the reconstructed Michalis complex (Fig. 4a), which show a very different conformation of the Glc in UDP-Glc compared with the ones observed for UDP-Gal and UDP-2F-Glc (Supplementary Fig. 9). The modeled MC shows that, as expected from the ternary complex structure, the side chain of the acceptor asparagine is not engaged in any interaction with a basic protein residue that could act as a general base in the reaction. However, the amino group of the amide side chain interacts with the α-phosphate of UDP, suggesting a pathway for acceptor deprotonation that is reminiscent of the one proposed for the closely related O-GlcNAc transferase (OGT), an inverting GT from the same GT family that shares the GT-B architecture[37,46,48].

QM/MM metadynamics simulations of the reaction mechanism showed that the reaction starting from the most common amide of Asn[0] involves a high energy barrier, which we attribute to a highly restrained nucleophilic attack stereochemistry. However, Asn[0] can tautomerize in the active site and adopt the imidic form, which exhibits

a more favorable configuration for the nucleophilic attack, resulting in a significant decrease of the reaction free energy barrier of the reaction. Our findings indicate that tautomerization of Asn residues is crucial for glycosylation in NGTs. Recently, asparagine tautomerization has also been invoked to mediate or initiate glycosylation reactions in other carbohydrate-active enzymes, such as protein *O*-fucosyltransferase 1 (POFUT1)[49], cellulase Cel45A[50] and GH85 endo-β-glucosaminidases[51].

Another significant discovery from our study is the identification of the α-phosphate as the base responsible for deprotonating the acceptor Asn. This differs from the proposed mechanism for OSTs, which involves a twisted amide promoted by the binding of the amide group to metal-coordinated acidic residues[15], but is similar to the one proposed[37] (and confirmed by QM/MM simulations[46]) for *O*-glycosylation catalyzed by OGT. Additionally, we found that NGTs recognize Thr[+2] differently from OSTs[15,26], suggesting that the two enzymes might employ distinct mechanisms to achieve *N*-glycosylation.

NGTs are not only interesting for their intriguing structural and mechanistic aspects, but also for their biotechnological properties as

possible therapeutic targets for treating infectious diseases. In terms of biotechnological applications, NGTs have been targeted to synthesize more homogeneous *N*-glycans in combination with endo-β-N-acetylglucosaminidases (ENGases) and synthetic oligosaccharides. This approach has proven useful in enhancing the stability of therapeutic peptides such as glucagon-like peptide 1[52]. Moreover, NGTs have been employed alongside subsequent GTs to produce custom-made glycoproteins and glycoprotein-based nanomaterials with potential biomedical applications[53]. However, it is worth noting that NGTs primarily add glucose onto Asn residues, while GlcNAc residues are more desirable in certain cases for *N*-glycans in proteins from eukaryotes. Our structural and mechanistic insights might provide a potential solution to facilitate the engineering of NGTs to achieve *N*-GlcNAcylation. Furthermore, our work also opens up possibilities for synthetizing structure-based inhibitors to treat diseases caused by non-typeable *H. influenzae* or other Gram-negative bacteria, considering the high similarity between the active sites of NGTs.

In summary, our experimental and computational work reveals the molecular basis of UDP-Glc and peptide recognition by *Aa*NGT and propose that the enzyme follows a concerted single-displacement mechanism using one UDP phosphate group as general base. Additionally, we highlight the importance of the tautomeric form of Asn acceptor residues as a necessary step for glycosylation. This study exemplifies how GTs employ different strategies to activate less reactive nucleophilic groups such as Asn residues to achieve glycosylation.

## Methods

### Cloning and purification of *Aa*NGT
The DNA sequence encoding amino acid residues of the *Aa*NGT protein (aa 1-621) was codon optimized and synthesized by GenScript (USA) for expression in *E. coli* cells. This construct was a gift from Dr. Min Chen at Shandong University[30]. The construct was subcloned in pMALC2x, rendering the vector pMALC2x-10Hist-PP-*Aa*NGT. The plasmid contained a sequence encoding a 10xHis tag and a PreScission protease (PP) cleavage site between the maltose binding protein (MBP) and the protein of interest. *Aa*NGT mutants (R177A, H214A, D215A, and R177A-H214A-D215A) were generated by GenScript via site-directed mutagenesis using the vector pMALC2x-10HistTAG-PP-*Aa*NGT.

The plasmids were transformed into BL21 (DE3) Gold cells and colonies were selected on LB/Agar plates containing 100 μg/ml of ampicillin. The cultures were grown at 37 °C in 2XTY medium (16 g/l tryptone, 10 g/l yeast extract powder, 5 g/l NaCl, pH 7.5) containing 100 μg/ml of ampicillin. When the optical density of the cultures reached 0.6, they were induced with 1 mM IPTG (isopropyl β-D-thiogalactoside) and incubated at 18 °C for 18 h. The cells were harvested by centrifugation at 17,700 × g at 4 °C for 10 min and resuspended in buffer A (25 mM TRIS pH 7.5, 300 mM NaCl, 10 mM Imidazole). The protein was then loaded onto a His-Trap column (GE Healthcare) and eluted with an imidazole gradient from 10 mM to 500 mM (buffer B: 25 mM TRIS pH 7.5, 300 mM NaCl, 500 mM imidazole). Buffer exchange to buffer C (25 mM TRIS pH 7.5, 150 mM NaCl) was carried out using a HiPrep 26/10 Desalting Column (GE Healthcare). To remove the MBP, the protein PP was added to the fusion protein and the mix was incubated at 4 °C for 18 h. The cleavage of the fused protein was confirmed with a SDS-page gel and the MBP was removed with a His-Trap Column. The fractions containing the *Aa*NGT protein were then concentrated to ~2.5 mL using centrifugal filter units of 30,000 MWCO cutoff (Millipore). Subsequently, gel filtration chromatography was carried out using Superdex 75 XK26/60 column (Cytiva) in buffer C to further remove impurities. The protein was then concentrated once again and protein concentration was measured by absorbance at 280 nm and by using its theoretical extinction coefficient ($\varepsilon_{280nm}^{AaNGT} = 67270\,M^{-1}cm^{-1}$). Note that a similar extinction coefficient was used to determine the concentration for the mutants. If the protein was used to produce crystals, it was buffer exchanged to buffer D (25 mM TRIS pH 7.5) prior to get concentrated. Additionally, the purity of the proteins was confirmed by SDS-PAGE.

### Isothermal titration microcalorimetry
ITC was used to characterize the interaction of *Aa*NGT with different ligands. All experiments were carried out in an Auto-iTC200 (Microcal, GE Healthcare) at 25 °C with *Aa*NGT at 100 μM and UDP at 1.2 mM, UDP-2F-Glc/UDP-5S-Glc at 4 mM and UDP-Gal at 3.5 mM in 25 mM TRIS pH 7.5 150 mM NaCl. All the experiments were repeated at least two times independently with similar results, and one representative plot with the derived dissociation constant $K_D$ and standard error of fitting for each experiment is shown (Fig. 1d, e, and Supplementary Fig. 2 and Table 2). Data integration, correction, and analysis were carried out in Origin 7 (Microcal) and the data were fitted to a one-site equilibrium-binding model. Stoichiometry (n) of binding in all cases was ~1:1.

### Kinetic analysis
Enzyme kinetics for the wild type and mutants were determined using the UDP-Glo luminescence assays (Promega). Initially, the wild type and mutants were tested in reactions containing 500 nM of enzyme in 25 mM Tris pH 7.5, 150 mM NaCl, 500 μM of peptide FG<u>N</u>WTT and 500 μM of UDP-Glc. To determine the $K_m$ of UDP-Glc, the reaction contained 500 μM of the peptide and UDP-Glc concentrations ranging from 5 μM to 1 mM. To measure the $K_m$ of the peptide, the reaction contained 500 μM UDP-Glc and the peptide concentrations ranging from 5 μM to 1 mM. To assess the different nucleotides (UDP-Glc, UDP-Gal, UDP-2F-Glc, and UDP-5S-Glc), reactions contained 500 nM of the wild-type enzyme in 25 mM Tris pH 7.5, 150 mM NaCl, and 500 μM of the sugar nucleotides and the peptide FG<u>N</u>WTT. The reactions were incubated for 30 min at 37 °C and stopped using 5 μl of UDP-detection reagent at a 1:1 ratio in a white, opaque 384-well plate and incubated in the dark for 1 h at room temperature before measuring with a Synergy HT (Biotek). To estimate the amount of UDP produced in the glycosyltransferase reaction, we created a UDP standard curve. The values were corrected against the UDP-Glc (or other sugar nucleotides when applicable) hydrolysis and were fit to a non-linear Michaelis–Menten program in GraphPad Prism 8 software from which the $K_m$, $k_{cat}$, and $V_{max}$ along with their standard errors were obtained. All experiments were performed in duplicate.

### Solid-phase peptide synthesis
The peptide was synthesized by stepwise microwave-assisted solid-phase synthesis on a Liberty Blue synthesizer using the Fmoc strategy on Rink Amide MBHA resin (0.1 mmol). All other Fmoc amino acids (5.0 equiv.) were automatically coupled using oxyma pure/DIC (*N,N*′-diisopropylcarbodiimide). The peptide was then released from the resin, and all acid-sensitive sidechain protecting groups were simultaneously removed using TFA 95%, TIS (triisopropylsilane) 2.5% and $H_2O$ 2.5%, followed by precipitation with cold diethyl ether. The crude products were purified by HPLC on a Phenomenex Luna C18(2) column (10 μm, 250 mm × 21.2 mm) and a dual absorbance detector, with a flow rate of 10 mL/min.

### Peptide preparation
The peptide used in this work was dissolved at 100 mM in buffer 25 mM Tris-HCl pH 7.5. The pH of each solution was measured with pH strips and when needed adjusted to pH 7–8 through the addition of 0.1–5 μL of 2 M NaOH.

### Crystallization
Crystals of the *Aa*NGT complexes were grown by sitting drop experiments at 18 °C by mixing 0.4 μl of protein solution (17 mg/mL *Aa*NGT and 5 mM ligands in buffer D "25 mM TRIS pH 7.5") with an equal volume of a reservoir solution. The crystals for the different complexes

were obtained in different conditions. The crystals for the *Aa*NGT-UDP-peptide complex were obtained in a 22% polyacrylic acid 5100 sodium salt, 100 mM HEPES sodium salt pH 7. We further soaked these crystals with the same condition saturated with peptide (a tiny little amount of solid was dissolved in the drop with the crystals) and 25 mM UDP for 30 min before flash freezing in a cryoprotectant solution containing 20% ethylene glycol. The crystals for the *Aa*NGT-UDP-Gal complex were obtained in 0.1 M magnesium chloride, 0.1 M Na HEPES pH 7.5, 10% (w/v) PEG 4000 solution. These crystals were further soaked 5 min with 37.5 mM UDP-Gal prepared in buffer D and flash frozen in the cryoprotectant solution. The crystals for the *Aa*NGT-UDP-2F-Glc complex were obtained in a 0.1 M calcium acetate, 0.1 M sodium acetate pH 4.5, 10% (w/v) PEG 4000 solution. We further soaked these crystals with 30 mM UDP-2F-Glc for 5 min before flash freezing in the cryoprotectant solution.

## Structure determination and refinement
Diffraction data for the three crystals of *Aa*NGT were collected on synchrotron beamlines I03 of the Diamond Light Source (Harwell Science and Innovation Campus, Oxfordshire, UK) and XALOC beamline at the ALBA synchrotron (Barcelona, Spain) at a wavelength of 0.97 Å and a temperature of 100 K. XDS[54] and CCP4 software packages[55] were used for data processing and scaling. Relevant statistics are presented in Table 1. Molecular replacement with Phaser[55] and PDB entry 3Q3E as a template was used to solve the crystal structures. Initial phases were further improved by cycles of manual model building in Coot[56] and restrained refinement with REFMAC5[55]. Further rounds of model building in Coot with TLS refinement in REFMAC5 were performed for all complexes. The crystal structures were validated with PROCHECK and model statistics are presented in Table 1. The Ramachandran plot for the *Aa*NGT-UDP-Gal complex shows that 95.1%, 4.6%, 0.1%, and 0.2% of the amino acids are in most favored, allowed, generously allowed and disallowed regions, respectively. The Ramachandran plot for the *Aa*NGT-UDP-2F-Glc complex shows that 90.6%, 8.8%, 0.4%, and 0.2% of the amino acids are in most favored, allowed, generously allowed and disallowed regions, respectively. The Ramachandran plot for the *Aa*NGT-UDP-peptide complex shows that 90.1%, 8.6%, 1.0%, and 0.3% of the amino acids are in most favored, allowed, generously allowed and disallowed regions, respectively. The asymmetric unit of the $P2_12_12_1$ crystals contained two molecules of *Aa*NGT.

## NMR
$^1$H and $^{19}$F NMR spectra were recorded at 400 and 376 MHz using a Bruker AVANCE 400 Plus Nanobay in chloroform-d or deuterium oxide. $^{13}$C and $^{31}$P NMR spectra were recorded at 101 and 162 MHz with the same instruments in chloroform-d or deuterium oxide. Chemical shifts are given in ppm (δ) and referenced to tetramethylsilane or to the internal solvent signal used as an internal standard. Assignments in the NMR spectra were made by first-order analysis of the spectra, and were supported by 1H−1H COSY, 1H−13C HMQC correlation results. High-resolution mass spectrometry was performed on a Waters Synapt G2-S HDMS spectrometer. Unless otherwise stated, all the commercially available solvents and reagents were purchased from FUJIFILM Wako Pure Chemical Corporation and Merck KGaA without further purification. During purification by silica gel column chromatography, the absorbances of all fractions were measured at 262 nm using a JASCO UV-2075 Plus detector.

## Synthesis of UDP-2F-Glc and UDP-5S-Glc
Synthesis of UDP-2F-Glc (7) and UDP-5S-Glc (14) and their precursor species (1-6 and 8-13; see Supplementary Figs. 3 and 4) was performed as described below.

1,3,4,6-Tetra-*O*-acetyl-β-D-mannopyranose **1** (200 mg, 574 µmol) was dissolved in 1,4-dioxane (5.00 mL) and cooled at 0 °C. The solution was added (diethylamino)sulfur Trifluoride (220 µL, 1.68 mmol) and the mixture was then heated at 100 °C by irradiating microwave for 5 min. The mixture was diluted with dichloromethane and washed with ice water, aqueous sodium hydrogen carbonate and brine, dried over anhydrous sodium sulfate, filtered, and evaporated. The residue was purified by silica gel chromatography with 5:1 to 3:1 (v/v) hexane:ethyl acetate to give compound **2** (1,3,4,6-Tetra-*O*-acetyl-2-deoxy-2-fluoro-β-D-glucopyranose; 182 mg, 91%); $^1$H NMR (400 MHz, CDCl₃) δ 5.77 (dd, 1H, $J_{1,2}$ = 8.1 Hz, $J_{1,F}$ = 3.1 Hz, H-1), 5.37 (dt, 1H, $J_{3,F}$ = 14.4 Hz, $J_{2,3}$ = 9.1 Hz, $J_{3,4}$ = 9.2 Hz, H-3), 5.06 (t, 1H, H-4), 4.45 (ddd, 1H, $J_{2,3}$ = 9.1 Hz, $J_{2,F}$ = 50.8 Hz, H-2), 4.29 (dd, 1H, H-6a, $J_{6a,6b}$ = 12.6 Hz), 4.14 (dd, 1H, H-6b), 3.86 (ddd, 1H, H-5), 2.18 (s, 3H, Ac), 2.09 (s, 3H, Ac), 2.08 (s, 3H, Ac), 2.04 (s, 3H, Ac); $^{19}$F NMR (376 MHz, CDCl₃) δ −200.9 (ddd, $J_{F,2}$ = 53.7 Hz, $J_{F,1}$ = 2.9 Hz, $J_{F,3}$ = 15.2 Hz).

Compound **2** (101 mg, 289 µmol) was dissolved in DMF (900 µL) added ammonium carbonate (233 mg, 2.42 mmol). The mixture was then stirred at 20 °C for 8 h. The mixture was diluted with ethyl acetate and washed with water and brine, dried over anhydrous sodium sulfate, filtered, and evaporated. The residue was purified by silica gel chromatography with 6:1 to 3:2 (v/v) hexane:ethyl acetate to give compound **3** (3,4,6-Tri-*O*-acetyl-2-deoxy-2-fluoro-α/β-D-glucopyranose; 74.1 mg, 83%); $^1$H NMR (400 MHz, CDCl₃) δ 5.59 (dd, 1H, $J_{3,F}$ = 12.1 Hz, $J_{3,4}$ = 9.5 Hz, H-3α), 5.48 (d, 1H, $J_{1,2}$ = 3.6 Hz, H-1α), 5.32 (dt, 1H, $J_{3,F}$ = 14.1 Hz, $J_{2,3}$ = 5.4 Hz, $J_{3,4}$ = 5.4 Hz, H-3β), 5.04 (m, 2H, H-4α, H-4β), 4.92 (dd, 1H, $J_{1,2}$ = 7.6 Hz, $J_{1,F}$ = 2.7 Hz, H-1β), 4.52 (ddd, 1H, $J_{2,F}$ = 49.4 Hz, $J_{2,3}$ = 9.5 Hz, H-2α), 4.30−4.09 (m, 6H, H-2β, H-5α, H-6αa, H-6αb, H-6βa, H-6βb), 3.94 (s, 1H, OH-1β), 3.77 (dq, 1H, H-5β), 3.55 (s, 1H, OH-1α), 2.09 (s, 3H, Ac), 2.09 (s, 3H, Ac), 2.08 (s, 6H, Ac), 2.04 (s, 3H, Ac), 2.04 (s, 3H, Ac).

Compound **3** (42.7 mg, 139 µmol) and DMAP (102 mg, 834 µmol) were co-evaporated with dry toluene. The mixture was dissolved in dry dichloromethane (1.30 mL) and cooled at −10 °C. The solution was then added diphenyl chlorophosphate (86.0 µL, 416 µmol), and the mixture was stirred at 20 °C for 30 min. The mixture was diluted with dichloromethane and washed with ice water, aqueous 1 M HCl, aqueous sodium hydrogen carbonate and brine, dried over anhydrous sodium sulfate, filtered, and evaporated. The residue was purified by silica gel chromatography with 5:1 to 2:1 (v/v) hexane:ethyl acetate to give compound **4** (3,4,6-Tri-*O*-acetyl-2-deoxy-2-fluoro-α-D-glucopyranosyl diphenylphosphate; 29.0 mg, 39%,); $^1$H NMR (400 MHz, CDCl₃) δ 7.40-7.22 (m, 10H, Ph), 6.14 (dd, 1H, $J_{1,2}$ = 3.6 Hz, $J_{1,F}$ = 6.6 Hz, H-1), 5.56 (dt, 1H, $J_{3,F}$ = 11.8 Hz, $J_{3,4}$ = 9.6 Hz, H-3), 5.07 (t, 1H, $J_{4,5}$ = 10.0 Hz, H-4), 4.62 (dq, 1H, $J_{2,F}$ = 48.4 Hz, $J_{2,3}$ = 9.6 Hz, H-2), 4.15 (dd, 1H, $J_{6a,6b}$ = 12.7 Hz, $J_{5,6b}$ = 3.9 Hz, H-6b), 4.00 (m, 1H, H-5), 3.79 (dd, 1H, $J_{5,6a}$ = 2.1 Hz, H-6a), 2.01 (s, 3H, Ac), 2.05 (s, 3H, Ac), 2.03 (s, 3H, Ac); $^{13}$C[$^1$H] NMR (101 MHz, CDCl₃) δ 170.5, 170.00, 169.5 (*C*OCH₃×3), 130.1, 130.0, 125.9, 125.9, 120.5, 120.4, 120.2, 120.2 (Ph), 94.7 (dd, C-1, $J_{1,P}$ = 5.4 Hz, $J_{1,F}$ = 22.6 Hz), 86.6 (dd, C-2, $J_{2,P}$ = 8.4 Hz, $J_{2,F}$ = 198 Hz), 70.0 (d, C-3, $J_{3,F}$ = 19.5 Hz), 69.6 (C-5), 67.0 (d, C-4, $J_{4,F}$ = 7.4 Hz), 60.9 (C-6), 20.8, 20.7, 20.6 (CO*C*H₃).

Compound **4** (28.4 mg, 52.6 µmol) was dissolved in ethyl acetate/methanol (1:1, 1.00 mL) and the flask was filled with argon. The solution was added Platinum oxide (IV) (15.2 mg) and the mixture was then stirred at 20 °C for 4 h in hydrogen atmosphere. The mixture was filtered through celite, and the filtrate was added triethylamine (16.0 µL, 115 µmol) and stirred for 30 min. The solution was evaporated to give a syrup including compound **5** (Bis (triethylammonium) 3,4,6-tri-*O*-acetyl-2-deoxy-2-fluoro-α-D-glucopyranosyl phosphate).

The mixture containing compound **5** (31.0 mg, 52.6 µmol) was co-evaporated with pyridine and dissolved in pyridine (500 µL). The solution was added to UMP-morpholidate (54.4 mg, 79.2 µmol), which evaporated with pyiridine, in the reaction flask. Moreover, 1*H*-tetrazole (13.4 mg, 191 µmol) was co-evaporated with pyridine and dissolved in pyridine (500 µL). The solution was transferred to the reaction flask by

a syringe and the mixture was then stirred at room temperature for 39 h. The mixture was evaporated and the residue was purified by silica gel chromatography with 9:1 to 6:1 (v/v/v) acetonitrile:water. The absorbance of each fraction was measured at 262 nm and the combined fractions were evaporated. The residue was purified by silica gel chromatography with 15:2:1 to 6:2:1 (v/v/v) ethyl acetate:methanol:water to give compound **6** (Uridine 5′-diphospho-3,4,6-tri-O-acetyl-2-deoxy-2-fluoro-α-D-glucopyranose; 25.7 mg, 70%); $^1$H NMR (400 MHz, D$_2$O) δ 7.97 (d, 1H, $J_{5'',6''}$ = 8.1 Hz, H-6″), 5.95–5.91 (m, 2H, H-1′, H-5″), 5.81 (dd, 1H, $J_{1,2}$ = 3.6 Hz, $J_{1,F}$ = 7.4 Hz, H-1), 5.43 (dt, 1H, $J_{3,F}$ = 12.1 Hz, $J_{2,3}$ = 9.4 Hz, $J_{3,4}$ = 9.4 Hz, H-3), 5.03 (t, 1H, H-4), 4.79–4.65 (m, 1H, H-2), 4.38–4.37 (m, 2H, H-2′, H-3′), 4.34 (s, 1H, H-5), 4.25–4.22 (m, 2H, H-4′, H-5′), 4.17–4.09 (m, 2H, H-6a, H-6b); $^{13}$C[$^1$H] NMR (101 MHz, D$_2$O) δ 171.4 (COCH$_3$), 170.9 (COCH$_3$), 170.6 (COCH$_3$), 164.0 (C-4″), 149.5 (C-2″), 139.5 (C-6″), 100.4 (C-1′), 89.9 (dd, $J_{1,P}$ = 5.2 Hz, $J_{1,F}$ = 22 Hz, C-1), 86.4 (C-5″), 84.5 (dd, $J_{2,P}$ = 8.6 Hz, $J_{2,F}$ = 194 Hz, C-2), 80.8 (d, $J_{4',P}$ = 9.3 Hz, C-4′), 71.8 (C-2′), 68.8 (d, $J_{3,F}$ = 19.4 Hz, C-3), 67.2 (C-3′), 65.8 (C-5), 65.4 (d, $J_{4,F}$ = 7.5 Hz, C-4), 62.7 (d, $J_{5',F}$ = 5.3 Hz, C-5′), 59.3 (C-6), 18.0 (COCH$_3$), 17.9 (COCH$_3$), 17.8 (COCH$_3$); $^{19}$F NMR (376 MHz, D$_2$O) δ −200.9 (ddd, $J_{F,2}$ = 53.7 Hz, $J_{F,1}$ = 2.9 Hz, $J_{F,3}$ = 15.2 Hz); $^{31}$P NMR (162 MHz, D$_2$O) δ −11.59 (d, $J_{P,P}$ = 17.5 Hz), −13.89 (d, $J_{P,P}$ = 15.7 Hz).

Compound **6** (18.9 mg, 27.2 μmol) was dissolved in triethylamine/methanol/water (1:2:2, 5.0 mL) and stirred at −20 °C for 12 h. The solution was then evaporated and lyophilized to give compound **7** (Uridine 5′-diphospho-2-deoxy-2-fluoro-α-D-glucopyranose triethylammonium salt; 16.9 mg, 93%); $^1$H NMR (400 MHz, D$_2$O) δ 7.88 (d, 1H, $J_{5'',6''}$ = 8.0 Hz, H-6″), 5.94–5.90 (m, 2H, H-1′, H-5″), 5.74 (dd, 1H, $J_{1,2}$ = 3.6 Hz, $J_{1F}$ = 7.4 Hz, H-1), 4.46–4.29 (dd, 1H, H-2), 4.31 (m, 2H, H-2′, H-3′), 4.23–4.11 (m, 2H, H-4′, H-5′), 3.97 (dt, 1H, $J_{2,3}$ = 3.6 Hz, $J_{3,4}$ = 3.6 Hz, $J_{3,F}$ = 12.8 Hz, H-3), 3.86 (dq, 1H, H-5), 3.80 (dd, 1H, $J_{6a,6b}$ = 12.7 Hz, $J_{5,6b}$ = 3.9 Hz, H-6b), 3.72 (dd, 1H, $J_{5,6a}$ = 2.1 Hz, H-6a), 3.47 (t, 1H, H-4), 3.14 (q, 1.8H, (CH$_3$CH$_2$)$_3$N), 1.21 (t, 3H, (CH$_3$CH$_2$)$_3$N); $^{13}$C[$^1$H] NMR (101 MHz, D$_2$O) δ 167.8 (C-4″), 153.0 (C-2″), 141.5 (C-6″), 102.7 (C-5″), 92.6 (dd, C-1, $J_{1,P}$ = 5.5 Hz, $J_{1,F}$ = 22.6 Hz), 90.4–88.5 (dd, C-2, $J_{2,P}$ = 8.3 Hz, $J_{2,F}$ = 188 Hz), 88.5 (C-1′), 83.1 (d, C-4′, $J_{4'P}$ = 9.0 Hz), 73.7 (C-2′), 72.7 (C-5), 71.0 (d, C-3, $J_{3,F}$ = 17.3 Hz), 69.7 (C-3′), 68.6 (d, C-4, $J_{4,F}$ = 7.9 Hz), 65.0 (d, C-5′, $J_{5',P}$ = 5.4 Hz), 60.1 (C-6); $^{19}$F NMR (376 MHz, D$_2$O) δ −200.38 (dd, $J_{F,2}$ = 49.0 Hz, $J_{F,1}$ = 12.9 Hz); $^{31}$P NMR (162 MHz, D$_2$O) δ −11.71 (d, $J_{P,P}$ = 18.5 Hz), −13.45 (dd, $J_{P,P}$ = 18.7 Hz); HRMS (ESI/Q-TOF) m/z: [M − H]$^-$ Calcd for C$_{15}$H$_{22}$F$_1$N$_2$O$_{16}$P$_2^-$: 567.0434; Found: 567.0447.

5-Thio-D-glucopyranose **8** (29.7 mg, 151 μmol) was dissolved in pyridine (1.80 mL) and added acetic anhydride (900 μL, 9.52 mmol). The mixture was stirred at 20 °C for 4 h. The mixture was then evaporated to give compound **9** (1,2,3,4,6-Penta-O-acetyl-5-thio-α/β-D-glucopyranose; 61.6 mg, quant.); $^1$H NMR (400 MHz, CDCl$_3$) δ 6.15 (d, 1H, $J_{1α,2}$ = 3.2 Hz, H-1α), 5.89 (d, 1H, $J_{1β,2}$ = 8.6 Hz, H-1β), 5.44 (t, 1H, $J_{3α,4}$ = 10.2 Hz, H-3α), 5.39–5.27 (m, 3H, H-4α, H-2β, H-4β), 5.24 (dd, 1H, $J_{2α,3}$ = 10.2 Hz, H-2α), 5.11 (t, 1H, $J_{3β,2}$ = 9.0 Hz, $J_{3β,4}$ = 9.0 Hz, H-3β), 4.38 (dd, 1H, $J_{6b,6a}$ = 11.9 Hz, $J_{6b,5}$ = 5.6 Hz, H-6αb) 4.31 (dd, 1H, H-6βb), 4.15 (dd, 1H, $J_{6a,5}$ = 3.0 Hz, H-6βa), 4.06 (dd, 1H, H-6αb), 3.59 (dq, 1H, H-5α), 3.31 (dq, 1H, H-5β), 2.08 (s, 3H, Ac), 2.07 (s, 3H, Ac), 2.04 (s, 3H, Ac), 2.04 (s, 3H, Ac), 2.02 (s, 3H, Ac), 2.02 (s, 3H, Ac), 2.01 (s, 3H, Ac), 1.99 (s, 3H, Ac).

Compound **9** (48.8 mg, 120 μmol) was dissolved in DMF (500 μL) added hydrazine monohydrate (8.77 μL, 180 μmol) and acetic acid (10.2 μL, 179 μmol). The mixture was then stirred at 20 °C for 30 min. The mixture was diluted with ethyl acetate and washed with water, aqueous sodium hydrogen carbonate and brine, dried over anhydrous sodium sulfate, filtered, and evaporated. The residue was purified by silica gel chromatography with 3:1 to 1:1 (v/v) hexane:ethyl acetate to give compound **10** (2,3,4,6-Tetra-O-acetyl-5-thio-α/β-D-glucopyranose; 30.6 mg, 70%); $^1$H NMR δ (400 MHz, CDCl$_3$) 5.53 (dt, 1H, $J_{2α,3}$ = 9.5 Hz, $J_{3α,4}$ = 9.5 Hz, H-3α), 5.29 (dd, 1H, $J_{4α,5}$ = 10.8 Hz, H-4α), 5.25–5.20 (m, 2H, H-2β, H-4β), 5.16–5.13 (m, 2H, H-1α, H-2α), 5.14 (t, 1H, H-3β), 4.86 (d, 1H, $J_{1β,2}$ = 9.0 Hz, H-1β), 4.37 (dd, 1H, $J_{6b,6a}$ = 12.0 Hz, $J_{6b,5}$ = 4.9 Hz, H-

6αb), 4.28 (dd, 1H, H-6βb), 4.13–4.09 (m, 1H, H-6βa), 4.07 (dd, 1H, $J_{6a,5}$ = 3.2 Hz, H-6αa), 3.68 (dq, 1H, H-5α), 3.23 (dq, 1H, H-5β), 2.07-2.00 (s, 24H, Ac).

Compound **10** (30.6 mg, 84.0 μmol) and DMAP (60.0 mg, 491 μmol) were co-evaporated with dry toluene. The mixture was dissolved in dry dichloromethane (1.30 mL) and cooled at −10 °C. The solution was then added diphenyl chlorophosphate (51.0 μL, 247 μmol), and the mixture was stirred at 20 °C for 30 min. The mixture was diluted with dichloromethane and washed with ice water, aqueous 1 M HCl, aqueous sodium hydrogen carbonate and brine, dried over anhydrous sodium sulfate, filtered, and evaporated. The residue was purified by silica gel chromatography with 4:1 to 2:1 (v/v) hexane:ethyl acetate to give compound **11** (2,3,4,6-Tetra-O-acetyl-5-thio-α-D-glucopyranosyl diphenylphosphate; 27.4 mg, 55%); $^1$H NMR (400 MHz, CDCl$_3$) δ 7.39–7.21 (m, 10H, Ph), 5.87 (dd, 1H, $J_{1,2}$ = 2.9 Hz, H-1), 5.49 (t, 1H, H-3, $J_{3,4}$ = 9.9 Hz), 5.30 (dd, 1H, H-4, $J_{4,5}$ = 10.9 Hz), 5.15 (dt, 1H, H-2, $J_{2,3}$ = 9.9 Hz), 4.32 (dd, 1H, H-6b, $J_{6b,6a}$ = 12.2 Hz, $J_{6b,5}$ = 4.7 Hz), 3.89 (dd, 1H, H-6a, $J_{6a,5}$ = 2.9 Hz), 3.41 (dq, 1H, H-5); $^{13}$C[$^1$H] NMR (101 MHz, CDCl$_3$) δ 170.5, 169.8, 169.7, 169.5 (COCH$_3$×4), 130.0, 125.9, 120.7, 120.7, 120.4, 120.4 (Ph), 78.0 (d, $J_{1,P}$ = 7.2 Hz, C-1), 73.8 (d, $J_{2,P}$ = 6.1 Hz, C-2), 71.4 (C-4), 70.20 (C-3), 60.69 (C-6), 39.73 (C-5), 20.7, 20.6, 20.6, 20.3 (COCH$_3$×4).

Compound **11** (27.4 mg, 45.9 μmol) was dissolved in ethyl acetate/methanol (1:1, 1.00 mL) and the flask was filled with argon. The solution was added Platinum oxide (IV) (17.1 mg) and the mixture was then stirred at 20 °C for 4 h in hydrogen atmosphere. The mixture was filtered through celite, and the filtrate was added triethylamine (16.0 μL, 115 μmol) and stirred for 20 min. The solution was evaporated to give a syrup including compound **12** (Bis (triethylammonium) 2,3,4,6-tetra-O-acetyl-5-thio-α-D-glucopyranosyl phosphate).

The mixture containing compound **12** (29.7 mg, 45.9 μmol) was co-evaporated with pyridine and dissolved in pyridine (500 μ). The solution was added to UMP-morpholidate (47.2 mg, 68.7 μmol), which evaporated with pyiridine, in the reaction flask. Moreover, 1H-tetrazole (11.3 mg, 161 μmol) was co-evaporated with pyridine and dissolved in pyridine (500 μL). The solution was transferred to the reaction flask by a syringe and the mixture was then stirred at room temperature for 42 h. The mixture was evaporated and the residue was purified by silica gel chromatography with 9:1 to 6:1 (v/v/v) acetonitrile:water. The absorbance of each fraction was measured at 262 nm and the combined fractions were evaporated. The residue was purified by silica gel chromatography with 9:2:1 to 7:2:1 (v/v/v) ethyl acetate:methanol:water to give compound **13** (Uridine 5′-diphospho-2,3,4,6-tetra-O-acetyl-5-thio-α-D-glucopyranose triethylammonium salt; 18.3 mg, 53%); $^1$H NMR δ (D$_2$O, 400 MHz) 7.97 (d, 1H, H-6″), 5.96–5.93 (m, 2H, H-1′, H-5″), 5.55 (dd, 1H, $J_{1,2}$ = 2.9 Hz, $J_{1,P}$ = 7.0 Hz, H-1), 5.40 (t, 1H, $J_{3,4}$ = 9.7 Hz, H-3), 5.26 (dd, 1H, $J_{4,5}$ = 10.8 Hz, H-4), 5.16 (dt, 1H, H-2), 4.46 (dd, 1H, $J_{6b,6a}$ = 12.4 Hz, $J_{6b,5}$ = 3.7 Hz, H-6b), 4.33–4.28 (m, 2H, H-2′, H-3′), 4.26–4.17 (m, 2H, H-4′, H-5′), 4.04 (dd, 1H, H-6a, $J_{6a,5}$ = 2.6 Hz), 3.72 (dt, 1H, H-5); $^{13}$C[$^1$H] NMR δ (101 MHz, D$_2$O) 173.7 (COCH$_3$), 172.9 (COCH$_3$), 172.8 (COCH$_3$), 166.2 (C-4″), 151.8 (C-2″), 141.9 (C-6″), 102.6 (C-5″), 88.6 (C-1′), 83.0 (d, $J_{4'P}$ = 9.2 Hz, C-4′), 74.6 (d, $J_{1,P}$ = 9.2 Hz, C-1), 74.3 (d, $J_{2,P}$ = 7.1 Hz, C-2), 73.9 (C-2′), 72.0 (C-4), 71.4 (C-3), 69.5 (C-3′), 65.0 (d, $J_{5',P}$ = 5.3 Hz, C-5′), 61.3 (C-6), 38.3 (C-5), 20.3 (COCH$_3$), 20.0 (COCH$_3$), 20.0 (COCH$_3$); $^{31}$P NMR (162 MHz, D$_2$O) δ −11.59 (d, $J_{P,P}$ = 16.4 Hz), −13.40 (d, $J_{P,P}$ = 10.3 Hz).

Compound **13** (29.6 mg, 39.4 μmol) was dissolved in triethylamine/methanol/water (1:2:2, 8.0 mL) and stirred at −20 °C for 12 h. The solution was then evaporated and lyophilized to give compound **14** (Uridine 5′-diphospho-5-thio-α-D-glucopyranose; 22.9 mg, 85%); $^1$H NMR (400 MHz, D$_2$O) δ 8.04 (d, 1H, $J_{5'',6''}$ = 8.1 Hz, H-6″), 5.94–5.91 (m, 2H, H-1′, H-5″), 5.54 (d, 1H, $J_{1,2}$ = 5.0 Hz, H-1), 4.41 (dt, 1H, H-2), 4.35 (dd, 1H, H-2′), 4.30 (t, 1H, H-3′, $J_{2',3'}$ = 4.3 Hz, $J_{2',1'}$ = 4.3 Hz), 4.21 (m, 1H, H-4′), 4.00 (dt, 1H, H-5′b), (m, 1H, H-5′a), 3.88–3.86 (m, 1H, H-6ab), 3.77 (t, 1H, $J_{2,3}$ = 9.6 Hz, $J_{3,4}$ = 9.6 Hz, H-3), 3.60 (t, 1H, $J_{4,5}$ = 10.0 Hz, H-4), 3.19–3.17

(m, 1H, H-5); $^{13}$C[$^1$H] NMR (100 MHz, D$_2$O) δ 166.3 (C-4″), 151.9 (C-2″), 102.6 (C-5″), 88.4 (C-1′), 84.0 (d, C-4′, $J_{4',P}$ = 8.7 Hz), 82.4 (C-2), 78.3 (d, C-1, $J_{1,P}$ = 3.5 Hz), 74.4 (C-3), 73.9 (C-2′), 71.4 (C-4), 70.0 (C-3′), 63.3 (C-5′, $J_{5',P}$ = 4.4 Hz), 59.5 (C-6), 43.9 (C-5); $^{31}$P NMR (162 MHz, D$_2$O) δ13.21 (d, $J_{P,P}$ = 19.7 Hz), 2.89 (s); HRMS (ESI/Q-TOF) m/z: [M − H]$^-$ Calcd for C$_{15}$H$_{23}$N$_3$O$_{16}$P$_2$S$_1^-$: 581.0249; Found: 581.0237.

ESI-high-resolution mass spectrometry spectra are shown for UDP-2F-Glu (compound **7**) and UDP-5S-Glc (compound **14**) (see Supplementary Fig. 10).

### Molecular dynamics simulations in explicit water

The X-ray structure obtained in this work (PDB 8P0Q), containing the enzyme and the acceptor FG**N**WTT peptide, was used in all simulations. In order to obtain a productive Michaelis complex, involving the donor UDP-glucose and the acceptor peptide, a docking procedure was needed, as the straightforward alignment of the crystal structures of the enzyme in complex with acceptor and donor (*Aa*NGT + peptide and *Aa*NGT + UDP-Gal or *Aa*NGT + UDP-Glc-2F, respectively) was not possible (the sugar moiety completely clashed with residues from the peptide). The structure was prepared for docking by removing subunit B, as well as the UDP molecules. The protonation of titratable residues (Asp, Glc, His) at pH 7 was decided by visual inspection as well as the online software MolProbity.

Docking of UDP-Glc into the structure of *Aa*NGT+peptide was performed with AutoDock Vina[57] and AutoDockTools 1.5.6. A group of selected residues near the active site were made flexible to better accommodate the substrate. Rotation of all the chemical bonds of UDP-Glc was allowed except for the dihedrals of the phosphates, to better mimic the phosphate configuration of UDP-GlcNAc of *O*-GlcNAc transferase (OGT)[58]. The values of the box size were taken as 32 x 30 x 26 Å, with the center of the box at 47.959, 62.228, 56.623 Å. Only one hit was obtained, with an affinity of −13.8 kcal/mol, which was used to perform molecular dynamics (MD) simulations.

The ternary complex obtained from the docking procedure was then prepared for classical MD simulations. The necessary files were obtained with the AmberTools suite[59]. UDP charges were parametrized with Gaussian09[60] and the antechamber program. The forcefields used for the protein, glucose, and UDP were F14SB[61], GLYCAM06[62], and GAFF[63], respectively. TIP3P was used for water molecules[64]. The enzyme was solvated in a box of dimensions 97.868 × 112.646 × 114.217 Å, resulting in a total of 32129 water molecules. The system was neutralized by adding 18 sodium ions.

Amber20 was used to perform the MD simulations[59]. The simulation procedure was as follows. First, the solvent was minimized without the protein and substrate, using positional restraints. Then, the whole system was minimized. Afterwards, the system was heated up to 300 K in a step-wise manner. The density of the system was taken to approximately 1 g/cm$^3$. Subsequently, MD in the NPT ensemble was performed, restraining the distance between the donor and the acceptor for 60 ns before the restraint was released. The simulation was enlarged up to 460 ns, from which the last 400 ns were taken as production run. The evolution of the protein and acceptor peptide RMSD is provided in Supplementary Fig. 5.

Additional MD simulations were performed considering Asn3$^0$ in its tautomeric (imidic acid) form, following the same protocol as the one described above. Smooth restraints were applied in the distance between the donor and acceptor as well as the dihedral angles of the donor sugar were restrained for the first 50 ns of the MD simulation, followed by 150 ns of unrestrained MD, which was taken as production run. The system was found to be stable, with the Asn3$^0$ side chain well oriented for nucleophilic attack for most of the run. The evolution of the protein and acceptor peptide RMSD is provided in Supplementary Fig. 11.

### QM/MM MD simulations

A representative snap-shot from the classical MD simulations in which the amide side chain interacts with the α-phosphate was selected to start the QM/MM MD simulations. All QM/MM simulations were performed using CP2K 9.1[65] interfaced with PLUMED 2.8[66]. The QM/MM simulations combined Born-Oppenheimer molecular dynamics at DFT level for the QM atoms, handled with the QM program QUICKSTEP[65,67], with classical MD for the rest of the system (simulated with the CP2K MM program FIST[65]). The QM region consisted of 44 atoms (counting capping atoms), including the donor glucose, two pyrophosphates, and the acceptor Asn (divided at the Cβ). The MM region included 106371 atoms. The boundary between the QM and MM regions was handled with the use of link H atoms via the Integrated Molecular Orbital Molecular Mechanics (IMOMM) method[68]. The QM region was enclosed inside a cell of 11.989 x 12.882 x 18.059 Å, and it was treated with the PBE functional[69], along with DFTD3 corrections[70], employing the GPW (Gaussian and plane-waves) scheme. GTH pseudopotentials[71] were used, and the Gaussian triple-ζ valence polarized (TZV2P) basis set. A plane-wave cut-off of 400 Ry was employed, together with a MD time step of 0.5 fs. The simulations were performed in the NVT ensemble, using a Nosé-Hoover thermostat to keep the temperature around 300 K. Non-bonded cut-off was 12 Å.

The initial snapshot from the classical MD was optimized using the conjugate gradient algorithm, then equilibrated for 5 ps before starting the metadynamics[44,72] simulations of the reaction the mechanism. The collective variable (CV) chosen was the difference between the Asn(N)-C1 and UDP-C1 distances. An upper wall was placed on distances related to the nucleophilic attack and deprotonation in order to limit the phase space accessible for the simulation (parabolic-type, with a force constant of 150 in internal PLUMED units). The Gaussian height was 1.0 kcal/mol, the width 0.1 Å and the deposition pace was set at 80 MD steps. The system reached the reaction products after having deposited 600 Gaussian functions (24 ps in terms of simulation time). However, the energy barrier was found to be very high (>50 kcal/mol), indicating that the reaction is not feasible. Thus, we stopped the simulation at this point. Alternative trials using different collective variables gave a similar negative result. Representative states of the reaction can be found in Supplementary Fig. 6.

An alternative enhanced sampling method (OPES) was also used to model the glycosylation reaction, within the QM/MM formalism. OPES[73] is a recently developed technique, related to metadynamics, that in its "explore" formulation is very useful for a fast inspection of possible reaction mechanisms. An QM/MM OPES simulation of the reaction was performed starting from the same structure as the previous QM/MM metadynamics simulation. The same collective variable including the distance difference between the main bonds to be broken/formed (C1-O$_P$ and N-C1) was used. Additionally, since we knew from the previous simulation that the α-phosphate abstracts the amide proton, we used another collective variable that includes the two distances involved in the proton transfer (N-H and H−O$_α$). The previously described upper wall on distances was used. The OPES parameters used were 25 kcal/mol for the energy barrier, and 80 steps for the MD pace (40 fs). Surprisingly, the simulations show that Asn3$^0$ rapidly undergoes tautomerization before performing the nucleophilic attack. Even though the simulation was only exploratory, it gave us the idea that Asn could react in its imidic form.

An unbiased QM/MM MD (7.2 ps) of the system with imidic Asn was performed to check its stability, before another QM/MM metadynamics simulation of the chemical reaction was launched. Two distances involving in the nucleophilic attack, N-C1 and leaving group departure (C1-O$_P$) were used as collective variables. The same upper wall on distances was used. The metadynamics simulations were performed using a Gaussian height of 1.2 kcal/mol and a width of 0.1 (CV

units) for both CVs, together with a pace of 80 MD steps (40 fs). A total of 2900 Gaussian functions were deposited, which in terms of real time amount to 116 ps. The simulation was stopped once the simulation crossed twice over the TS, as recommended for chemical reactions[74]. The location of the TS on the computed free energy landscape was further refined by committor analysis, using 30 independent replicas (18 reactants, 12 products).

To estimate the free energy required for the Asn to undergo tautomerization, we performed further QM/MM metadynamics simulations considering tautomerization via the α-phosphate (as observed serendipitously in a previous simulation). We used two collective variables; the difference in coordination numbers between $H-O_p$ and N-H (CV1) and the difference in coordination numbers between $O_{Asn}$-H and $H-O_p$ (CV2). The simulations used a Gaussian height of 0.8 kcal/mol, a width of 0.04 Å for both CVs, and a deposition pace of 100 MD steps (50 fs). The simulation run for a total of 166 ps, until recrossing over the transition state took place. The computed free energy barrier (29.5 kcal/mol) indicated that tautomerization via the α-phosphate is not particularly favored (it is more difficult than tautomerization of amides in water solution[75]). Therefore, we launched a second simulation to assess whether tautomerization with the participation of water molecules was more likely. To this end, a larger QM region, including two active site water molecules (Wat1 and Wat2) and the Lys440 side chain (coordinating the pyrophosphate groups) was used. The QM/MM metadynamics simulation was performed using two CVs: CV1 = CN(N-H) + CN($H_{Wat}$-$O_{Wat1}$) + CN($H_{Wat2}$-$O_{Wat2}$) and CV2 = CN($H-O_{Wat1}$) + CN($H_{Wat1}$-$O_{Wat2}$) + CN($H_{Wat2}$-O) (CN = coordination number). An upper wall was placed on distances related to the nucleophilic attack (not deprotonation) (parabolic-type, with a force constant of 150 in internal PLUMED units). The first CV is maximum when Asn is in its amide form, whereas the second one accounts for the imidic acid form. The simulation was performed using a Gaussian height of 1 kcal/mol and a width of 0.08 Å for both CVs, together with a deposition pace of 80 MD steps (40 fs). The simulation run for a total of 9.2 ps. In all cases, we used the following formula for coordination number (Eq. 1, see below):

$$CN_{ij} = \frac{1 - \left(\frac{r_{ij}}{r_0}\right)^n}{1 - \left(\frac{r_{ij}}{r_0}\right)^m} \qquad (1)$$

Where $CN_{ij}$ is the coordination number between atoms i and j, $r$ the distance between them, n are set at 6 and 12 respectively, and $r_0$ is 2.1 for bonds not involving H atoms and 1.2 for bonds that do involve them.

### Reporting summary
Further information on research design is available in the Nature Portfolio Reporting Summary linked to this article.

## Data availability
The crystal structures of the *Aa*NGT-UDP-Gal, *Aa*NGT-UDP-2F-Glc, and *Aa*NGT-UDP-peptide complexes were deposited at the RCSB PDB with accession code, 8P0O, 8P0P, and 8P0Q, respectively. Previously published PDB structures used in this study are available under the accession codes: 3Q3E, 3Q3I, 3Q3H, and 7Y4I. Other data are available from the corresponding author upon request. The kinetics and ITC data generated in this study are provided in the source data file. Source data are provided as a Source Data file. Source data are provided with this paper.

## Code availability
Data files of the classical MD simulation and QM/MM metadynamics simulations have been deposited in Zenodo (https://doi.org/10.5281/zenodo.8081487).

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

## Acknowledgements

We thank the Diamond Light Source (Oxford, UK) synchrotron beamline I03 (experiment number MX20229-7) and the ALBA (Barcelona, Spain) synchrotron beamline XALOC. We thank the Aragón Foundation for Research & Development (ARAID), the Agency for Management of University and Research Grants of Catalonia (AGAUR, 2021-SGR-00680 to C.R.), the Agencia Estatal de Investigación (AEI; BFU2016-75633-P, PID2019-105451GB-I00 and PID2022–136362NB-I00 to R.H.-G., PID2021–127622OB-I00 to F.C., PID2020-118893GB-I00 and CEX2021-001202-M to C.R.), the European Research Council (ERC-2020-SyG-951231"Carbocentre" to C.R.) and Gobierno de Aragón (E34_R17, E35_17R, and LMP58_18) with FEDER (2014-2020) funds for "Building Europe from Aragón" for financial support. B.P. acknowledges AGAUR for a PhD scholarship (2020 FI_B 00423), and M.G. acknowledges Marie Skłodowska-Curie fellowship (grant agreement No 101034288) for financial support. The authors would like to thank the technical support provided by the Barcelona Supercomputing Center (BSC) and Red Nacional de Supercomputación (RES; application BCV-2023-2-0016) for computer resources at MareNostrum IV and CTE-Power supercomputers.

## Author contributions

R.H.-G. designed the crystallization construct and solved the crystal structures. J.M.-L and B.V. performed the expression and purification of all proteins, the ITC experiments, and crystallized the complexes. A.G.-G. and V.T. performed the enzyme kinetics experiments. M.G. and I.C. synthetized the peptide. B.P. performed the computational experiments. A.M. and S.M. performed the synthesis of the UDP-Glc mimetics. R.H.-G. and C.R. wrote the article with mainly the contribution of F.C. and A.M. All authors read and approved the final manuscript.

## Competing interests

The authors declare no competing interests.
