## [Peer Review File · Nature Communications]

Reviewers' Comments:

Reviewer #1:

Remarks to the Author:

The crystal structures of AaNGT in complex with a poor donor substrate (UDP-Gal) and a donor mimic (UDP-2F-Glc), as well as the enzyme in complex with a peptide acceptor (FGNWTT) and UDP are reported. The work provides mechanistic insights into NGT function by identifying amino acids involved in substrate recognition and simulations. The results generally support the authors' conclusions. The work provides an important foundation for increasing our understanding of the N-glycosyltransferases. However, major and minor comments would need to be addressed to improve the quality of the manuscript.

Major comments:

Line 125-137. The authors performed the isothermal titration calorimetry (ITC) experiments with AaNGT and UDP-Glc to measure their KD value and thermodynamic parameters (Fig 1d-e and Supplementary Table 2). This doesn't make sense, because the AaNGT could also catalyze the hydrolysis of UDP-Glc and this hydrolysis reaction would cause heat released or absorbed in the solution.

According to the crystallography table, the structures are of lower resolution since the CC1/2 at high resolution is around 0.4. The cutoff for the highest resolution shell should have a CC1/2 above 0.5. Therefore, the data must be rescaled to render a CC1/2 above 0.5 in the highest resolution shell. This will decrease the current resolution, but it will reflect better the resolution of the structure. Furthermore, the Rwork and Rfree for the structure of AaNGT and UDP-2F-Glc complex was refined to be 20.2% and 27%. The gap between Rwork and Rfree is close to 7%, suggesting the data has been overfitting. The authors should re-refine this structure and make sure the gap between Rwork and Rfree is around or less than 5%.

Lines 279-288 and 376-379. Not only for the OGT, the α -phosphate is presumed to deprotonate the hydroxyl group of the sugar acceptor Ser/Thr. Consider reinforcing the argument that the α -phosphate of the nucleotide-sugar could serve as the general base by also mention that a similar substrate-assisted catalysis mechanism was proposed for the plant POFUT SPINDY (PMID: 36456586).

Minor comments:

There are several typos in Supplementary Table 2, Supplementary Figure 2, and Supplementary Figure 4. The ligands, instead of 'UDP-Glu, UDP-2F-Glu and UDP-5S-Glu', should be UDP-Glc, UDP-2F-Glc and UDP-5S-Glc.

Sentence in lines 161-163 does not make much sense. "This indicates not only a highly similar GT-B fold...". Rewrite this sentence.

Left bottom panel in Figure 3a, the labels are confusing. Both '0' and '-1' are placed close to the side chain of F1. Isn't F1 at position '-2'?

Sentence in lines 204-208 is quite confusing. The authors should make it clear that the residues belong to either the peptide or AaNGT.

Line 381. Delete the second 'from'.

Lines 535-539. The Ramachandran plot parameters should be included in Table 1.

Line 821. Two accession codes mentioned here are the same, which needs to be corrected.

Reviewer #2:

Remarks to the Author:

Hurtado-Guerrero and co-workers report multidisciplinary studies on the asparagine-linked glycosylation mediated by N-glycosyltransferase enzymes (NGTs). There remains a lack of understanding as to how AdNGTs recognize substrates and inhibitors (exemplified by UDP-Gal and UDP-Glc) and achieve glycosylation. Although the crystal structures of AdNGT complexed to UDP-Gal, and UDP-2F-Glu, and UDP-FGNNWTT were characterized, comparison of these structures did not provide a useful insight into revealing the NGT catalytic mechanism due to steric clash between the sugar moieties and peptide. Thus, the authors utilized computational approaches, namely MD simulations and subsequent QM/MM MD metadynamics simulations to clarify a plausible reaction pathway. In passing, they found that the imidic Asn3 converted from the original amide form is a key species that enhances the N-glycosylation reaction via an SN2 reaction. The proposed mechanism also differs from the previous findings in that UDP phosphate group serves as general base. The paper is well written and deals with an important and challenging problem both on the experimental and computational sides. I recommend it for publication after a minor revision that addresses the following points.

Comments:

- Line 72 and 94, the numbers (0, +1, +2, +3) that indicate the positions of the peptide residues suddenly appear without explanation. The readers who are not familiar with this system would be confused. The authors should mention their definition. The bottom left of Fig. 2a may be comprehensive.
- Fig. 3b, it is difficult to distinguish between the three structures.
- Regarding the reaction pathway starting from the imidic tautomer, the authors indicated that the activation free energy was calculated to be 24.9 kcal/mol lying more than 20 kcal/mol below that for the reaction initiated by the amide amido tautomer. The authors should think about the possibility that the tautomerization requires a large activation energy. It would be convincing if they could evaluate the free energy barrier for this process. Also, they should check the stability of the system with the imidic tautomer by performing additional classic MD simulations.

Reviewer #3:

Remarks to the Author:

This study by Piniello and colleagues describes a thorough structural, kinetic, and computational investigation into the reaction mechanism of soluble bacterial asparagine glycosyltransferases (NGTs). These enzymes are of interest for their role in pathogenesis and their potential as biotechnological tools. Similar to oligosaccharyltransferases (OSTs), the catalytic mechanism of NGTs is of particular interest because the amide side chain of the acceptor asparagine is unreactive in its planar conformation, and it is unclear how the N is activated by these enzymes to perform nucleophilic attack at C1 of the donor sugar. As noted in the manuscript, a twisted amide hypothesis has been proposed for the OSTs, but no viable mechanism has been proposed for the NGTs.

Noteworthy results of the current study are:

- 1) New crystal structures with various substrates, mimetics, and inhibitors, especially the ternary complex which provides first insight into binding of the peptide substrate. The ternary complex also provides a basis for modelling the Michaelis complex.
- 2) Observation in the QM/MM calculations of spontaneous tautomerization of the amide side chain of the acceptor asparagine, suggesting a catalytic mechanism might proceed via this imidic form of asparagine.
- 3) Further QM/MM calculations that support a concerted one-step SN2 reaction mechanism involving the imidic form of the Asn side chain.

This work is novel and of considerable interest to the field. It can be expected to stimulate further experimental investigations into the reaction mechanism of this and related enzymes. It will also provide a basis for engineering novel substrate specificities into this important biotechnological

tool.

To the degree that I am able to assess it, the methodology used is sound and the conclusions are supported by the observations. I cannot extend this statement to the QM/MM calculations as this is beyond my expertise.

The manuscript is also exceptionally well written and clear to understand. I congratulate the authors.

Overall the manuscript submitted by Piniello and colleagues represents a ground breaking study into the biochemistry of glycosyltransferases. I wish to contribute only a few points that the authors may consider when revising the manuscript.

- 1) The free energy barrier calculated for the proposed reaction mechanism (24.9 kcal/mol) seems slightly on the high side for a glycosyltransfer reaction. Perhaps the authors can compare this with other values and make a comment as to whether this agrees with the observed kinetic parameters.
- 2) I would certainly be interested if the authors are able to report a free energy barrier for tautomerization of the asparagine side chain.
- 3) There is a typo on page 11, line 233. D125A should be D215A

Yours sincerely,
Tim Keys

REVIEWER COMMENTS

Reviewer #1 (Remarks to the Author):

The crystal structures of AaNGT in complex with a poor donor substrate (UDP-Gal) and a donor mimic (UDP-2F-Glc), as well as the enzyme in complex with a peptide acceptor (FGNWTT) and UDP are reported. The work provides mechanistic insights into NGT function by identifying amino acids involved in substrate recognition and simulations. The results generally support the authors' conclusions. The work provides an important foundation for increasing our understanding of the N-glycosyltransferases. However, major and minor comments would need to be addressed to improve the quality of the manuscript.

Response#: Thank you for your comments.

Major

comments:

- Line 125-137. The authors performed the isothermal titration calorimetry (ITC) experiments with AaNGT and UDP-Glc to measure their KD value and thermodynamic parameters (Fig 1d-e and Supplementary Table 2). This doesn't make sense, because the AaNGT could also catalyze the hydrolysis of UDP-Glc and this hydrolysis reaction would cause heat released or absorbed in the solution.

Response#: We agree with the reviewers and hydrolysis might take place during the ITC experiment with UDP-Glc.

Action#: In the revised version of our manuscript, we have removed the ITC for UDP-Glc and have updated Figures 1d and 1e.

- According to the crystallography table, the structures are of lower resolution since the CC1/2 at high resolution is around 0.4. The cutoff for the highest resolution shell should have a CC1/2 above 0.5. Therefore, the data must be rescaled to render a CC1/2 above 0.5 in the highest resolution shell. This will decrease the current resolution, but it will reflect better the resolution of the structure.

Response#: The reviewer raised a point that has been discussed in different publications. CC values range from 1 to -1 for perfectly correlated versus anticorrelated data, but for adequately indexed data, these indicators should range from near 1 for highly precise data to near 0 for very imprecise data. It is justified to include data below 0.5 depending on how many observations contributed to it. For our three data sets, we have enough observations (see our reflections in Table 1), a completeness of 100% and redundancy ranging from 6.6 to 8.1 to include data with a CC1/2 below 0.5. The significance of this value can be assessed by Student's test (e.g. $CC > 0.3$ is significant at $p = 0.01$ for $n > 100$;

CC>0.08 is significant at $p=0.01$ for $n>1000$) [Karplus and Diederichs, 2012]. It was empirically shown that the inclusion of data with a CC1/2 value between 0.1 and 0.2, $R_{meas} \sim 450\%$ and $\langle I/\sigma \rangle_{mrgd} \sim 0.3$ led to an improved refined model [Karplus and Diederichs, 2012]. Other studies have shown the same scenario, which are cited in the last reference below.

- Karplus PA, Diederichs K. Assessing and maximizing data quality in macromolecular crystallography. *Curr Opin Struct Biol.* 2015 Oct;34:60-8. doi: 10.1016/j.sbi.2015.07.003. Epub 2015 Jul 24. PMID: 26209821; PMCID: PMC4684713

- CC* - Linking crystallographic model and data quality. Video recorded at SBGrid/NE-CAT workshop 2014;

- CC1/2 - Karplus PA, Diederichs K. Linking Crystallographic Model and Data Quality. *Science.* 2012. doi: 10.1126/science.121823.

Due to that, we prefer to follow the above recommendations about CC1/2 which agrees with our cutoff for the highest resolution shell.

Action#: No action taken.

Furthermore, the R_{work} and R_{free} for the structure of AaNGT and UDP-2F-Glc complex was refined to be 20.2% and 27%. The gap between R_{work} and R_{free} is close to 7%, suggesting the data has been overfitting. The authors should re-refine this structure and make sure the gap between R_{work} and R_{free} is around or less than 5%.

Response#: Thank you for bringing this point to our attention.

In response to the reviewer's comment, we have made significant improvements to our current model in Coot and have performed TLS refinement. As a result, we have successfully reduced the gap to below 5%, as suggested.

Action#: Specifically, we have improved the R_{work}/R_{free} values, which are 0.197/0.245. Furthermore, we have provided clarification that the structure containing UDP-2F-Glc has also undergone TLS refinement, similar to the other structures that were previously refined. This information has been mentioned in the Methods section as well. We have also updated the values for the Ramachandran plot, B-factors and RMS deviations in Table 1. Finally, we have uploaded the modified PDB file to the PDB server, ensuring that the latest version is now available.

Lines 279-288 and 376-379. Not only for the OGT, the α -phosphate is presumed to deprotonate the hydroxyl group of the sugar acceptor Ser/Thr. Consider reinforcing the argument that the α -phosphate of the nucleotide-sugar could serve as the general base by also mention that a similar substrate-assisted catalysis mechanism was proposed for the plant POFUT SPINDY (PMID: 36456586).

Response#: Thank you for mentioning the SPY example.

Action#: In response to your comment, we have included the SPY example in the revised version of our manuscript. This example further supports our argument that the α -phosphate can act as a general base. Additionally, we have strengthened the evidence for the α -phosphate's role as a catalytic base by mentioning that its pKa is approximately 6.5, which is consistent with its catalytic role. Furthermore, we have included references in the revised version that demonstrate this pKa through NMR and computational studies. We have incorporated these comments in the revised version of our manuscript.

Minor comments:
There are several typos in Supplementary Table 2, Supplementary Figure 2, and Supplementary Figure 4. The ligands, instead of 'UDP-Glu, UDP-2F-Glu and UDP-5S-Glu', should be UDP-Glc, UDP-2F-Glc and UDP-5S-Glc.

Response#: Thank you for bringing these typos.

Action#: These typos are now fixed in the revised version of our manuscript.

Sentence in lines 161-163 does not make much sense. "This indicates not only a highly similar GT-B fold...". Rewrite this sentence.

Response#: We agree with the reviewer that that particular sentence did not make much sense.

Action#: We have removed that particular sentence because it did not add anything new with respect to the previous sentence.

Left bottom panel in Figure 3a, the labels are confusing. Both '0' and '-1' are placed close to the side chain of F1. Isn't F1 at position '-2'?

Response#: We agree with the reviewer's observation.

Action#: In response to this feedback, we have correctly numbered these positions in the figure.

Sentence in lines 204-208 is quite confusing. The authors should make it clear that the residues belong to either the peptide or AaNGT.

Response#: We acknowledge the reviewer's observation and agree with the comment.

Action#: In order to provide clarity regarding the distinction between residues belonging to the peptide and those belonging to the protein, we have added the consonant "p" as superscript to all the residues of the peptide along the manuscript. This ensures that it is evident which residues belong to the peptide and not the protein.

Line 381. Delete the second 'from'.

Response#: Thank you for finding this mistake.

Action#: We have removed the second "from".

Lines 535-539. The Ramachandran plot parameters should be included in Table 1.

Response#: The Ramachandran plot parameters according to the instructions in NCOMMS should be located in Methods and not in the table.

Action#: No action taken.

Line 821. Two accession codes mentioned here are the same, which needs to be corrected.

Response#: Thank you for finding this mistake.

Action#: We have added the right pdb codes.

Reviewer #2 (Remarks to the Author):

Hurtado-Guerrero and co-workers report multidisciplinary studies on the asparagine-linked glycosylation mediated by N-glycosyltransferase enzymes (NGTs). There remains a lack of understanding as to how AdNGTs recognize substrates and inhibitors (exemplified by UDP-Gal and UDP-Glc) and achieve glycosylation. Although the crystal structures of AdNGT complexed to UDP-Gal, and UDP-2F-Glu, and UDP-FGNNWTT were characterized, comparison of these structures did not provide a useful insight into revealing the NGT catalytic mechanism due to steric clash between the sugar moieties and peptide. Thus, the authors utilized computational approaches, namely MD simulations and subsequent QM/MM MD metadynamics simulations to clarify a plausible reaction pathway. In passing, they found that the imidic Asn3 converted from the original amide form is a key species that enhances the N-glycosylation reaction via an SN2 reaction. The proposed mechanism also differs from the previous findings in that UDP phosphate group serves as general base. The paper is well written and deals with an important and challenging problem both on the experimental and computational sides. I recommend it for publication after a minor revision that addresses the following points.

Response#: Thank you very much for your comments.

Comments:

- Line 72 and 94, the numbers (0, +1, +2, +3) that indicate the positions of the peptide residues suddenly appear without explanation. The readers who are not familiar with this system would be confused. The authors should mention their definition. The bottom left of Fig. 2a may be comprehensive.

Response#: We agree with the reviewer that the numbering should be clarified in the introduction section.

Action#: We have added this nomenclature in Figure 1a and have also defined it in the introduction section.

- Fig. 3b, it is difficult to distinguish between the three structures.

Response#: We agree with the reviewer's observation regarding the figure.

Action#: In order to enhance the quality of the figure, we have made the following improvements. Firstly, we have reduced the size of the sticks representing the ligands to ensure better clarity and visual representation. Additionally, we have magnified the region of interest to focus on the specific area being discussed.

- Regarding the reaction pathway starting from the imidic tautomer, the authors indicated that the activation free energy was calculated to be 24.9 kcal/mol lying more than 20 kcal/mol below that for the reaction initiated by the amide amido tautomer. The authors should think about the possibility that the tautomerization requires a large activation energy. It would be convincing if they could evaluate the free energy barrier for this process.

Response#: We agree with the reviewer that it could be that tautomerization requires a large energy barrier and, therefore, it is necessary to assess it.

Action#: We have performed additional QM/MM metadynamics simulations to quantify the free energy barrier required for tautomerization of Asn3 (now Asn3^P), considering two scenarios: (1) tautomerization via the α -phosphate (as serendipitously observed in a QM/MM OPES simulation). (2) tautomerization via active site water molecules (in this case, the QM region was increased to include the closest two water molecules to the reactive asparagine, Asn3^P). The results obtained (new Figure S7) show that (2) has the lowest energy barrier (17.5 vs. 29.5 kcal/mol), thus we conclude that tautomerization preferentially occurs via active site water molecules. This is a situation similar to amide tautomerization in solution. Importantly, the energy barrier for tautomerization is lower than that of the glycosylation reaction (24.9), indicating that Asn tautomerization should not be rate-limiting.

The tautomerization mechanisms and their respective free energy landscapes have been added as new Figure S7 of Supporting Information (replacing the old Figure) and the results have been described in manuscript page 16:

*“Finally, we sought to elucidate the most likely mechanism of Asn tautomerization in the AaNGT active site. To this end, we performed QM/MM metadynamics simulations of the tautomerization process considering two possible scenarios: tautomerization mediated by the α -phosphate or tautomerization via active site water molecules (**Supplementary Fig. 7**). In both cases, two collective variables were used to drive the system from the amidic to the imidic form of the Asn3 side chain. Whereas tautomerization via the α -phosphate involves an energy barrier of 29.5 kcal/mol, the energy barrier for tautomerization via water molecules reduces to 17.5 kcal/mol when Asn3^P undergoes tautomerization via water molecules, thus it is not rate-limiting. This indicates that asparagine tautomerization in the active site is feasible and it is mediated by active site water molecules that are properly positioned for proton shuttle.”*

Also, they should check the stability of the system with the imidic tautomer by performing additional classic MD simulations.

Response#: We agree with the reviewer

Action#: We have performed classical molecular dynamics simulations of AaNGT in complex with the peptide in which Asn3^P is in its imidic acid form. The simulations (150 ns) show a stable active site, with Asn3^P in a reactive configuration. In particular, the N atom of Asn3^P remains close to the C1 atom of the donor sugar and the hydroxyl group points towards the α -phosphate. The results have been included in the new Figure S10.

Reviewer #3 (Remarks to the Author):

This study by Piniello and colleagues describes a thorough structural, kinetic, and computational investigation into the reaction mechanism of soluble bacterial asparagine glycosyltransferases (NGTs). These enzymes are of interest for their role in pathogenesis and their potential as biotechnological tools. Similar to oligosaccharyltransferases (OSTs), the catalytic mechanism of NGTs is of particular interest because the amide side chain of the acceptor asparagine is unreactive in its planar conformation, and it is unclear how the N is activated by these enzymes to perform nucleophilic attack at C1 of the donor sugar. As noted in the manuscript, a twisted amide hypothesis has been proposed for the OSTs, but no viable mechanism has been proposed for the NGTs.

Noteworthy results of the current study are:

- 1) New crystal structures with various substrates, mimetics, and inhibitors, especially the ternary complex which provides first insight into binding of the peptide substrate. The ternary complex also provides a basis for modelling the Michaelis complex.
- 2) Observation in the QM/MM calculations of spontaneous tautomerization of the amide side chain of the acceptor asparagine, suggesting a catalytic mechanism might proceed via this imidic form of asparagine.
- 3) Further QM/MM calculations that support a concerted one-step SN2 reaction mechanism involving the imidic form of the Asn side chain.

This work is novel and of considerable interest to the field. It can be expected to stimulate further experimental investigations into the reaction mechanism of this and related enzymes. It will also provide a basis for engineering novel substrate specificities into this important biotechnological tool.

To the degree that I am able to assess it, the methodology used is sound and the conclusions are supported by the observations. I cannot extend this statement to the QM/MM calculations as this is beyond my expertise.

The manuscript is also exceptionally well written and clear to understand. I congratulate the authors.

Response#: Thank you very much for your comments.

Overall the manuscript submitted by Piniello and colleagues represents a ground breaking study into the biochemistry of glycosyltransferases. I wish to contribute only a few points that the authors may consider when revising the manuscript.

- 1) The free energy barrier calculated for the proposed reaction mechanism (24.9 kcal/mol) seems slightly on the high side for a glycosyltransfer reaction. Perhaps

the authors can compare this with other values and make a comment as to whether this agrees with the observed kinetic parameters.

Response#: We agree with the reviewer that the energy barrier is a bit high for the reaction, as the kinetic parameters reported in this work would be compatible with a 18.4 kcal/mol barrier (applying transition state theory).

We think that this difference is probably due to an imperfect Michaelis complex, as it was obtained through docking and it is very possible that the position of the phosphates was not optimal for the reaction, which could have some impact on the computed energy barrier due to their direct involvement in the deprotonation step.

The computed energy barrier for NGT is similar to the one computed for the closely related OGT, another GT41 enzyme (23.5 kcal/mol, reference 42), which was also somewhat higher than the experimental value (20.7-20.9 kcal/mol).

We also would like to add that the comparison of the computed barrier with the experimental value is not always straightforward. On one hand, conversion of the experimentally measured rate to a free energy barrier using transition state theory includes several approximations, in particular the assumption that the transmission coefficient in the Eyring-Polanyi equation is equal to 1. The comparison also assumes that the chemical reaction is rate-limiting. In view of these approximations and assumptions, together with the results available for other GTs in the literature, we believe that the computed free energy barrier is in reasonable agreement with experiment.

Action#: We have commented the above aspects in the manuscript (page 16).

“The computed free energy barrier (24.9 kcal/mol) is still somewhat higher than the one estimated from the experimental rate constant (18.4 kcal/mol, assuming Transition State Theory),⁴¹ probably due to an imperfect position of the phosphate groups in the initial structures. However, it is similar to the one previously computed for OGT (23.5 kcal/mol).⁴² Most importantly, the free energy barrier is much reduced compared to the one obtained for the reaction via the amide form of Asn3^P (> 50 kcal/mol), indicating that the reaction occurs preferably via the imidic form of Asn3^P.”

2) I would certainly be interested if the authors are able to report a free energy barrier for tautomerization of the asparagine side chain.

Response#: We agree with the reviewer that it could be that tautomerization requires a large energy barrier and, therefore, it is necessary to assess it.

Action#: We have performed additional QM/MM metadynamics simulations to quantify the free energy barrier required for tautomerization of Asn3 (now Asn3^P), considering two scenarios: (1) tautomerization via the α -phosphate (as serendipitously observed in a QM/MM OPES simulation). (2) tautomerization via active site water molecules (in this case, the QM region was increased to include

the closest two water molecules to the reactive asparagine, Asn3^P). The results obtained (new Figure S7) show that (2) has the lowest energy barrier (17.5 vs. 29.5 kcal/mol), thus we conclude that tautomerization preferentially occurs via active site water molecules. This is a situation similar to amide tautomerization in solution. Importantly, the energy barrier for tautomerization is lower than that of the glycosylation reaction (24.9), indicating that Asn tautomerization should not be rate-limiting.

The tautomerization mechanisms and their respective free energy landscapes have been added as new Figure S7 of Supporting Information (replacing the old Figure) and the results have been described in manuscript page 16:

*“Finally, we sought to elucidate the most likely mechanism of Asn tautomerization in the AaNGT active site. To this end, we performed QM/MM metadynamics simulations of the tautomerization process considering two possible scenarios: tautomerization mediated by the α -phosphate or tautomerization via active site water molecules (**Supplementary Fig. 7**). In both cases, two collective variables were used to drive the system from the amidic to the imidic form of the Asn3 side chain. Whereas tautomerization via the α -phosphate involves an energy barrier of 29.5 kcal/mol, the energy barrier for tautomerization via water molecules reduces to 17.5 kcal/mol when Asn3^P undergoes tautomerization via water molecules, thus it is not rate-limiting. This indicates that asparagine tautomerization in the active site is feasible and it is mediated by active site water molecules that are properly positioned for proton shuttle.”*

3) There is a typo on page 11, line 233. D125A should be D215A

Response#: Thank you for finding this mistake.

Action#: This is now fixed in the revised version of our manuscript.

Reviewers' Comments:

Reviewer #1:

Remarks to the Author:

All my concerns with the previous submission are addressed in this version. In my opinion, the manuscript is now suitable for NCOMMS.

Reviewer #2:

Remarks to the Author:

The authors thoroughly addressed all of my questions raised in the first round of the review, demonstrating results of additional QM/MM MD and classical MD simulations. As such, I recommend acceptance of this manuscript for publication in NCOMMS.